# Myristoylated Neuronal Calcium Sensor-1 captures the preciliary vesicle at distal appendages

**Tomoharu Kanie[1,2]\*, Roy Ng[1], Keene L Abbott[1], Niaj Mohammad Tanvir[3], Esben Lorentzen[3], Olaf Pongs[4], Peter K Jackson[1]\***

[1]Baxter Laboratory, Department of Microbiology & Immunology and Department of Pathology, Stanford University, Stanford, United States; [2]Department of Cell Biology, University of Oklahoma Health Sciences Center, Oklahoma City, United States; [3]Department of Molecular Biology and Genetics, Aarhus University, Aarhus, Denmark; [4]Institute for Physiology, Center for Integrative Physiology and Molecular Medicine, Saarland University, Saarbrücken, Germany

**\*For correspondence:**
Tomoharu-Kanie@ouhsc.edu (TK);
pjackson@stanford.edu (PKJ)

**Competing interest:** The authors declare that no competing interests exist.

**Abstract** The primary cilium is a microtubule-based organelle that cycles through assembly and disassembly. In many cell types, formation of the cilium is initiated by recruitment of preciliary vesicles to the distal appendage of the mother centriole. However, the distal appendage mechanism that directly captures preciliary vesicles is yet to be identified. In an accompanying paper, we show that the distal appendage protein, CEP89, is important for the preciliary vesicle recruitment, but not for other steps of cilium formation (Kanie et al., 2025). The lack of a membrane-binding motif in CEP89 suggests that it may indirectly recruit preciliary vesicles via another binding partner. Here, we identify Neuronal Calcium Sensor-1 (NCS1) as a stoichiometric interactor of CEP89. NCS1 localizes to the position between CEP89 and the centriole-associated vesicle marker, RAB34, at the distal appendage. This localization was completely abolished in *CEP89* knockouts, suggesting that CEP89 recruits NCS1 to the distal appendage. Similar to *CEP89* knockouts, preciliary vesicle recruitment as well as subsequent cilium formation was perturbed in *NCS1* knockout cells. The ability of NCS1 to recruit the preciliary vesicle is dependent on its myristoylation motif and *NCS1* knockout cells expressing a myristoylation defective mutant failed to rescue the vesicle recruitment defect despite localizing properly to the centriole. In sum, our analysis reveals the first known mechanism for how the distal appendage recruits the preciliary vesicles.

## Editor's evaluation

The identification of NCS1 as a distal appendage protein that captures preciliary vesicles has fundamental implications for understanding the early steps of ciliary assembly, furthering also a broader understanding of NCS1. Prior to this work, studies of NCS1 were focused on its roles in neurotransmission, but now must be considered in a larger context. The investigators used a variety of state-of-the-art methodologies to arrive at compelling conclusions. This work will be of relevance to cell biologists, especially those studying ciliary assembly, as well as human geneticists with an interest in cilia-related pathologies and neurobiologists studying NCS1.

## Introduction

The primary cilium is an organelle that consists of the microtubule-based axoneme surrounded by the ciliary membrane, which accumulates specific membrane proteins (e.g., G-protein-coupled receptors)

to serve as a sensor for extracellular environmental cues (*Reiter and Leroux, 2017*). The cilium extends from the mother centriole, and cycles assembly and disassembly, as the cell needs to disassemble the cilium prior to mitosis (*Vorobjev and Chentsov Yu, 1982*) to allow the centrosome to function within the spindle pole during mitosis. The process of ciliary formation, described first by *Sorokin, 1962*; *Sorokin, 1968* has been classified into the extra- and intracellular pathways (*Molla-Herman et al., 2010*). In the extracellular pathway used, for example, by Mardin–Darby canine kidney cells, the mother centriole is believed to first dock to the plasma membrane before the extension of the axonemal microtubule as well as ciliary membrane (*Jewett et al., 2021*). In the intracellular pathway used, for example, by retinal pigment epithelia (RPE) cells (*Molla-Herman et al., 2010*) and fibroblasts (*Sorokin, 1962*; *Molla-Herman et al., 2010*), the first step of cilium formation is attachment of the small vesicles, or so-called preciliary vesicles (*Lu et al., 2015*), to the distal end of the mother centriole (*Sorokin, 1962*), or more specifically to the distal appendage (*Schmidt et al., 2012*; *Sillibourne et al., 2013*). The distal appendage is a ninefold blade-like structure attached to the distal end of the mother centriole (*Anderson, 1972*; *Bowler et al., 2019*; *Paintrand et al., 1992*). The preciliary vesicle recruitment is followed by the fusion of the small vesicles to form a large ciliary vesicle (*Lu et al., 2015*), removal of CP110 (*Lu et al., 2015*), which is believed to cap the distal end of the mother centriole (*Spektor et al., 2007*), and subsequent axonemal extension, which is mediated at least partially by intraflagellar transport (IFT) (*Craft et al., 2015*). While the distal appendage is indispensable for all those steps, how exactly the distal appendage controls these multiple processes is largely unknown.

To understand the molecular roles of the distal appendage, we first need to uncover its molecular composition and identify critical functions of distal appendage proteins. In an accompanying paper, we comprehensively analyzed all known distal appendage proteins to date and revealed that the Centrosomal Protein 89 (CEP89) is important for preciliary vesicle recruitment, but not for other processes organizing cilium formation (*Kanie et al., 2025*). Since CEP89 lacks apparent lipid-binding motifs, we hypothesized that an interacting partner of CEP89 may bind to preciliary vesicle directly. We sought to identify and understand the protein directly recruiting the preciliary vesicle.

## Results

### Discovery of Neuronal Calcium Sensor-1 as a stoichiometric interactor of CEP89

To identify interacting partners of CEP89, we performed tandem affinity purification and mass spectrometry (TAP-MS) (*Rigaut et al., 1999*). Localization and affinity purification (LAP) (*Cheeseman and Desai, 2005*) tagged CEP89 was expressed in RPE immortalized with human telomerase (RPE-hTERT), and CEP89 was immunoprecipitated first by Green Fluorescent Protein (GFP) antibody beads followed by a second affinity precipitation by S-protein beads. Final eluates were resolved by sodium dodecyl sulfate–polyacrylamide gel electrophoresis (SDS–PAGE) gel and analyzed by silver staining (*Figure 1A*) and mass spectrometry (*Figure 1B*). This analysis identified two stoichiometric interactors, Neuronal Calcium Sensor-1 (NCS1) and CEP15, consistent with the previous high-throughput proteome analyses, which identified both proteins as either CEP89 interactors (*Huttlin et al., 2021*) or neighbors (*Gupta et al., 2015*). CEP15 was previously named as C3ORF14, and we renamed it to CEP15 to reflect its function. Consistent with the TAP-MS data, endogenous NCS1 strongly co-immunoprecipitated with endogenous CEP89 (*Figure 1—figure supplement 1A*).

NCS1 is a member of NCS family proteins, which are characterized as containing calcium-binding EF-hand motifs as well as a myristoylation signal for N-terminal addition of myristate (*Burgoyne and Weiss, 2001*). NCS1 was first identified as Frequenin in *Drosophila*, a protein that can facilitate neurotransmitter release in neuromuscular junction (*Pongs et al., 1993*). Since then, numerous papers proposed models wherein NCS1 is involved in both presynaptic and postsynaptic functions (reviewed in *Dason et al., 2012*). However, how exactly NCS-1 regulates neuronal function is still not well understood. As described later in this paper, NCS-1 is expressed ubiquitously in various tissues, consistent with the previous report (*Gierke et al., 2004*). While previous studies reported the roles for NCS1 in cardiomyocytes (*Nakamura et al., 2011*) and adipocytes (*Ratai et al., 2019*), the function of NCS1 in non-neuronal cells remain enigmatic. A centrosomal role of NCS1 has never been described.

CEP89 binding to NCS1 required the N-terminal region (1–343 a.a.) (*Figure 1C, D*) in agreement with the structural model of NCS1-CEP89 predicted using Alphafold2 showing that the very N-terminal

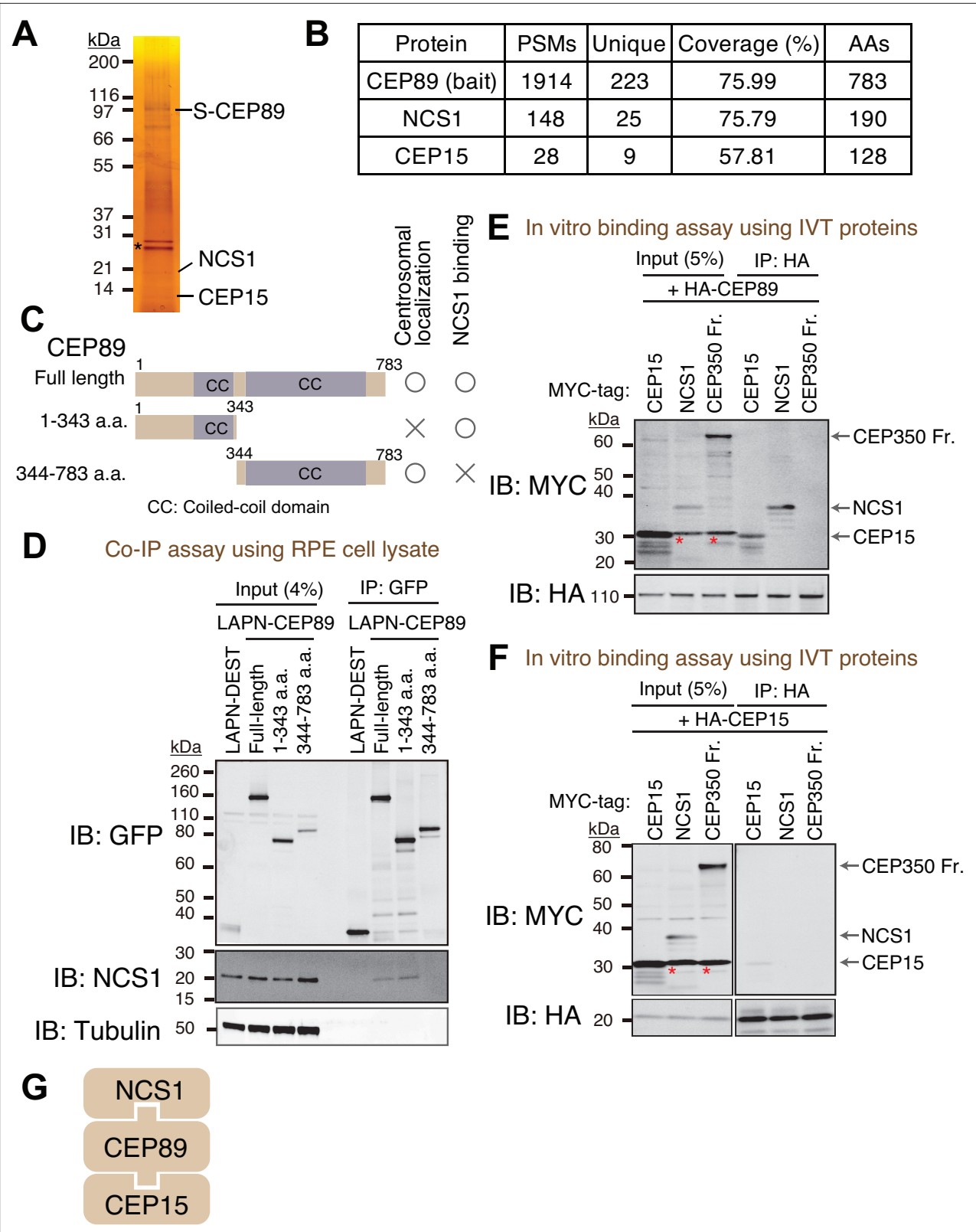

**Figure 1.** Identification of Neuronal Calcium Sensor-1 as a stoichiometric interactor of CEP89. (**A**) Silver staining of the eluate following tandem affinity purification of N-terminally LAP (EGFP-TEV cleavage site-S tag-PreScission cleavage site)-tagged CEP89 expressed in confluent retinal pigment epithelia (RPE) cells. The cell lysates were purified with GFP antibodies and S-protein beads, resolved by sodium dodecyl sulfate–polyacrylamide gel electrophoresis (SDS–PAGE) and visualized by silver staining. The bands corresponding to S-tagged CEP89 (S-CEP89), NCS1, and CEP15 are indicated.

*Figure 1 continued on next page*

*Figure 1 continued*

Molecular weights (kDa) estimated from a protein marker are indicated. Asterisk denotes a band corresponding to TEV protease used for tandem affinity purification. Uncropped image of silver staining can be found in *Figure 1—source data 1*. (**B**) Tabulation of peptide-spectrum matches (PSMs), unique peptide counts, coverage, and the length of the amino acids from the mass spectrometry analysis of the eluate shown in (**A**). Raw mass spectrometry data can be found in *Figure 1—source data 2*. (**C**) A cartoon depicting the region of CEP89 important for centrosomal localization or binding to NCS1. Localization data can be found in *Figure 1—figure supplement 1A*. (**D**) Immunoblot (IB) analysis of the eluates from a co-immunoprecipitation assay of the full length or the indicated fragments of N-terminally LAP-tagged CEP89 expressed in confluent RPE cells. The cell lysates were purified with GFP antibodies, resolved by SDS–PAGE and immunoblotted with the indicated antibodies. Molecular weights (kDa) estimated from a protein marker are indicated. The raw unedited blots can be found in *Figure 1—source data 3* and *Figure 1—source data 4*. Immunoblot (IB) analysis of the eluates from in vitro binding assay of the in vitro translated (IVT) N-terminally HA-tagged CEP89 (**E**) or CEP15 (**F**) and the indicated N-terminally MYC-tagged proteins. The in vitro translated proteins were mixed and captured by HA-agarose beads, resolved by SDS–PAGE and immunoblotted with the indicated antibodies. The CEP350 fragment (2470–2836 a.a.), which binds to FOP efficiently (*Figure 1—figure supplement 2A*; *Kanie et al., 2017*) serves as a negative control. Red asterisks indicate non-specific bands, which overlap with the MYC-tagged CEP15. Molecular weights (kDa) estimated from a protein marker are indicated. The raw unedited blots can be found in *Figure 1—source data 5*, *Figure 1—source data 6*, *Figure 1—source data 7*, and *Figure 1—source data 8*. (**G**) The order of binding for CEP89–NCS1–CEP15 interaction.

The online version of this article includes the following source data and figure supplement(s) for figure 1:

**Source data 1.** Uncropped image of silver staining of the tandem affinity purification analysis of CEP89 shown in *Figure 1A*.

**Source data 2.** Mass spectrometry analysis of tandem affinity purification of CEP89 shown in *Figure 1B*.

**Source data 3.** The original files of the full raw unedited blots shown in *Figure 1D*.

**Source data 4.** The uncropped blots with boxes that indicate the regions displayed in *Figure 1D*.

**Source data 5.** The original files of the full raw unedited blots shown in *Figure 1E*.

**Source data 6.** The uncropped blots with boxes that indicate the regions displayed in *Figure 1E*.

**Source data 7.** The original files of the full raw unedited blots shown in *Figure 1F*.

**Source data 8.** The uncropped blots with boxes that indicate the regions displayed in *Figure 1F*.

**Figure supplement 1.** Individual channels of the images shown in *Figure 1A*.

**Figure supplement 1—source data 1.** The original files of the full raw unedited blots shown in *Figure 1—figure supplement 1A*.

**Figure supplement 1—source data 2.** The uncropped blots with boxes that indicate the regions displayed in *Figure 1—figure supplement 1A*.

**Figure supplement 1—source data 3.** Immunofluorescence conditions in the experiment shown in *Figure 1—figure supplement 1B*.

**Figure supplement 2.** A negative control for the experiment shown in *Figure 1E, F*.

**Figure supplement 2—source data 1.** The original files of the full raw unedited blots shown in *Figure 1—figure supplement 2A*.

**Figure supplement 2—source data 2.** The uncropped blots with boxes that indicate the regions displayed in *Figure 1—figure supplement 2A*.

helix of CEP89 interacts with NCS1 (*Figure 2O*). The C-terminal portion of CEP89 (344–783 a.a.) is required for its centrosomal localization (*Figure 1—figure supplement 1B*), consistent with a previous report (*Sillibourne et al., 2013*). An in vitro binding assay using in vitro translated proteins revealed that HA-tagged CEP89 directly binds to MYC-tagged CEP15 and NCS1 (*Figure 1E*), whereas HA-CEP15 did not bind to NCS1 (*Figure 1F*). A negative control, CEP350 fragment (2470–2836 a.a.), which binds to its binding partner FGFR1OP (or FOP) efficiently (*Figure 1—figure supplement 2A*) as previously described (*Kanie et al., 2017*), did not bind to either HA-CEP89, nor HA-CEP15 (*Figure 1E, F*). Thus, CEP89 serves to bridge NCS1 and CEP15 (*Figure 1G*).

## NCS1 is recruited to the distal appendage by CEP89 and is positioned between CEP89 and the centriole-associated vesicle marker, RAB34

We next sought to determine the precise localization of NCS1 and CEP15. When observed via a wide-field microscopy, NCS1 localized to the mother centriole, marked by CEP170 (*Figure 2A*). Some cytoplasmic staining was also observed. Both centriolar and cytoplasmic staining was highly specific as the signal was strongly reduced in *NCS1* knockout cells (*Figure 2A*). The cytoplasmic localization of NCS1 is inconsistent with a previous study, where C-terminally Enhanced Yellow Fluorescent Protein (EYFP)-tagged NCS1 constitutively localized to membranous compartments (*O'Callaghan et al., 2002*). We tested if the difference in localization is due to the tagging. We also tested whether membrane binding, myristoylation motif, of NCS1 affects its localization by making the myristoylation defective mutant by converting the position 2 glycine to alanine (NCS1-G2A). N-terminally LAP (EGFP and S)-tagged wild-type or G2A mutant of NCS1 localized to mother centriole as well as cytoplasm,

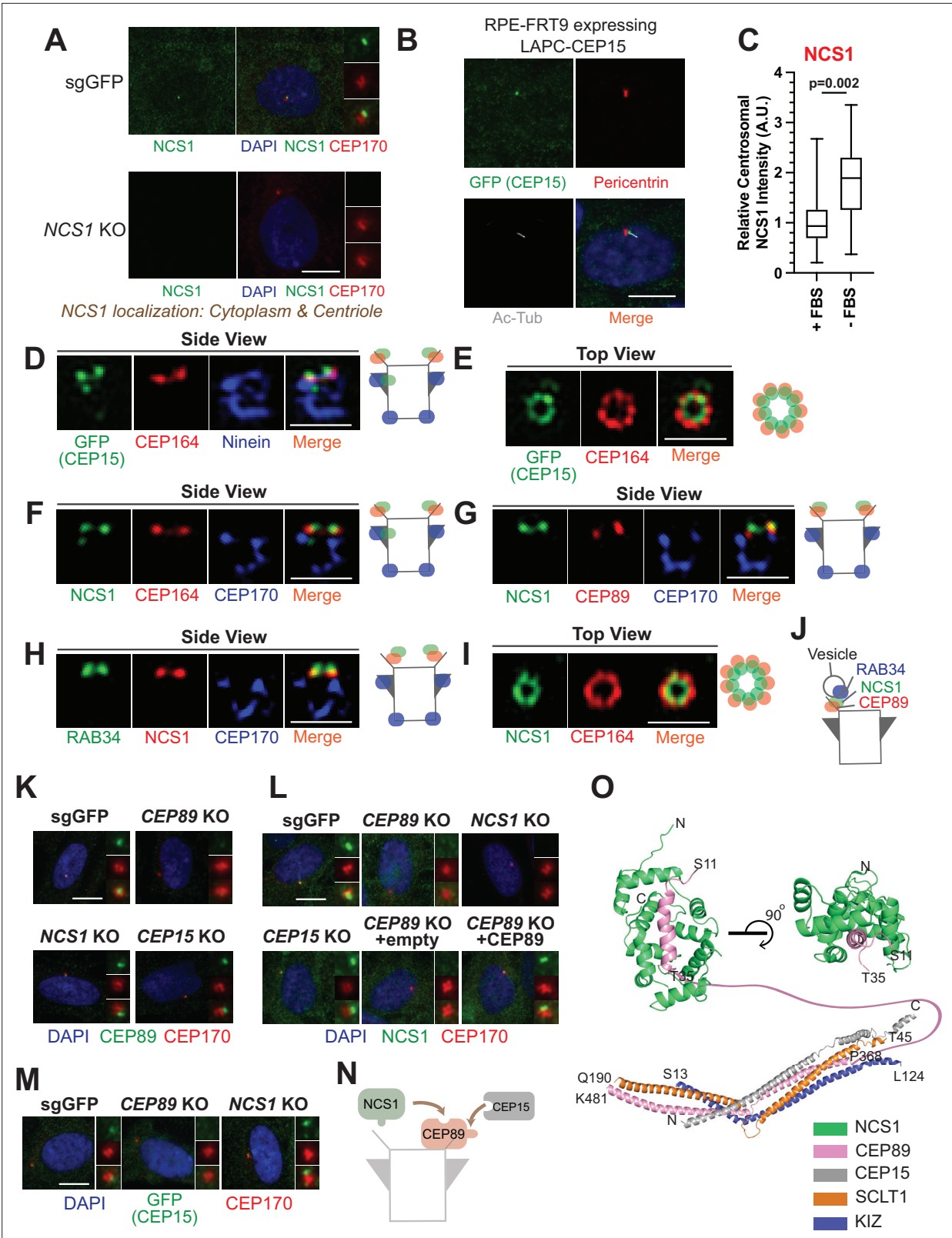

**Figure 2.** NCS1 is recruited to the distal appendage by CEP89. (**A**) Immunofluorescence images taken via wide-field microscopy. Control (sgGFP) or *NCS1* knockout retinal pigment epithelia (RPE) cells were serum starved for 24 hr, fixed, and stained with indicated antibodies. Insets at the right panels are the enlarged images of the mother centriole. Ac-Tub indicates acetylated α-tubulin. Scale bar: 10 μm. (**B**) Immunofluorescence images taken via wide-field microscopy. RPE cells expressing C-terminally LAP (LAPC)-tagged CEP15 were serum starved for 24 hr, fixed, and stained with

*Figure 2 continued on next page*

*Figure 2 continued*

indicated antibodies. Scale bar: 10 µm. (**C**) Box plots showing centrosomal signal intensity of NCS1. RPE cells were grown in fetal bovine serum (FBS)-containing media for 24 hr, and then grown in either FBS-containing media (+FBS) or serum-free media (−FBS) for an additional 24 hr. Cells were fixed and stained with NCS1 antibody. Centrosomal signal intensity of NCS1 was measured from fluorescence images using the method described in Materials and methods. A.U., arbitrary units. Data are combined from three replicates. Statistical significance was calculated from a nested *t*-test. The raw data, experimental conditions, and detailed statistics are available in *Figure 2—source data 3*. (**D–I**) Immunofluorescence images taken via 3D-structured illumination microscopy. Side view (D, F–H) or top view (**E, I**) is shown. RPE cells were either grown to confluent (**H**) or serum starved for 24 hr (**D–G, I**), fixed and stained with indicated antibodies. Each individual image is from a representative z-slice. Scale bar: 1 µm. CEP170: a marker of subdistal appendage and proximal end of the mother centriole. A cartoon at the right of each figure shows estimated positions of each protein at the mother centriole. (**J**) A cartoon depicting the localization of NCS1 relative to RAB34 and CEP89. NCS1 is sandwiched between RAB34 and CEP89. (**K–M**) Immunofluorescence images taken via wide-filed microscopy. Control (sgGFP) or indicated knockout RPE cells were serum starved for 24 hr, fixed, and stained with indicated antibodies. Scale bar: 10 µm. Insets at the right panels are the enlarged images of the mother centriole. Quantification data are available in *Figure 2—figure supplement 2A–C*. (**N**) A cartoon depicting the order of recruitment of the CEP89–NCS1–CEP15 complex. (**O**) Structural prediction of the NCS1/CEP89/CEP15/SCLT1/KIZ pentametric complex. (Bottom) AlphaFold2 prediction showing a tetrameric coiled-coil complex with each subunit displayed as cartoon representation and colored differently as indicated. (Top) Two perpendicular views of the structural prediction of the CEP89–NCS1 complex highlighting the N-terminal helix of CEP89 buried in a binding pocket of NCS1. Protein termini are labeled in the model and the residues between T35-P368 of CEP89 are indicated with a pink line.

The online version of this article includes the following source data and figure supplement(s) for figure 2:

**Source data 1.** Immunofluorescence conditions in the experiment shown in *Figure 2A*.

**Source data 2.** Immunofluorescence conditions in the experiment shown in *Figure 2B*.

**Source data 3.** Immunofluorescence conditions, raw image quantification data, and detailed statistics of the experiment shown in *Figure 2C*.

**Source data 4.** Immunofluorescence conditions in the experiment shown in *Figure 2D–I*.

**Source data 5.** Immunofluorescence conditions in the experiment shown in *Figure 2K*.

**Source data 6.** Immunofluorescence conditions in the experiment shown in *Figure 2L*.

**Source data 7.** Immunofluorescence conditions in the experiment shown in *Figure 2M*.

**Figure supplement 1.** Localization of GFP-tagged NCS1.

**Figure supplement 1—source data 1.** Immunofluorescence conditions in the experiment shown in *Figure 2—figure supplement 1*.

**Figure supplement 2.** Quantification data and immunoblot related to *Figure 2*.

**Figure supplement 2—source data 1.** Immunofluorescence conditions and raw quantification data of the experiment shown in *Figure 2—figure supplement 2A*.

**Figure supplement 2—source data 2.** Immunofluorescence conditions and raw quantification data of the experiment shown in *Figure 2—figure supplement 2B*.

**Figure supplement 2—source data 3.** Immunofluorescence conditions and raw quantification data of the experiment shown in *Figure 2—figure supplement 2C*.

**Figure supplement 2—source data 4.** The original files of the full raw unedited blots shown in *Figure 2—figure supplement 2D*.

**Figure supplement 2—source data 5.** The uncropped blots with boxes that indicate the regions displayed in *Figure 2—figure supplement 2D*.

**Figure supplement 2—source data 6.** The original files of the full raw unedited blots shown in *Figure 2—figure supplement 2E*.

**Figure supplement 2—source data 7.** The uncropped blots with boxes that indicate the regions displayed in *Figure 2—figure supplement 2E*.

**Figure supplement 3.** The predicted structure of the NCS1/CEP89/CEP15/SCLT1/KIZ pentameric complex.

**Figure supplement 4.** Localization of distal appendage protein in *NCS1* knockouts.

**Figure supplement 4—source data 1.** Immunofluorescence conditions, raw quantification data, and detailed statistics of the experiment shown in *Figure 2—figure supplement 4A*.

**Figure supplement 4—source data 2.** Immunofluorescence conditions, raw quantification data, and detailed statistics of the experiment shown in *Figure 2—figure supplement 4B*.

**Figure supplement 4—source data 3.** Immunofluorescence conditions and raw quantification data of the experiment shown in *Figure 2—figure supplement 4C*.

**Figure supplement 4—source data 4.** Immunofluorescence conditions and raw quantification data of the experiment shown in *Figure 2—figure supplement 4D*.

---

similar to endogenous NCS1 (*Figure 2—figure supplement 1A, B*). Consistent with the previous paper (*O'Callaghan et al., 2002*), C-terminally LAP-tagged wild-type NCS1 localized to membrane compartments, such as plasma membrane and endoplasmic reticulum, whereas the myristoylation defective mutant (G2A) diffusely localized to cytoplasm (*Figure 2—figure supplement 1C, D*). This

result suggests that C-terminal tagging of NCS1 changes its localization potentially via exposing the myristoylation motif of NCS1, and endogenous NCS1 may sequester its myristoylation motif to allow localization to cytoplasm. A small amount of nuclear localization observed in LAP-tagged NCS1 likely derives from LAP tagging (*Figure 2—figure supplement 1*), as the endogenous NCS1 did not localize to nucleus (*Figure 2A*). Similar to NCS1, C-terminally LAP-tagged CEP15 localized to the location between acetylated tubulin, a cilium marker, and Pericentrin, a centrosome marker, suggesting that it also localizes specifically to the mother centriole (*Figure 2B*). NCS1 localization to the mother centriole was enhanced upon serum deprivation (*Figure 2C*), a condition that induces cilium formation in RPE cells, much like several other distal appendage proteins (see Figure 1D of *Kanie et al., 2025*). When observed via 3D-structured illumination microscopy with a resolution twice as high as a diffraction limited microscopy (*Wu and Shroff, 2018*), C-terminally LAP (EGFP-S)-tagged CEP15 localized to a position slightly distal to the distal appendage protein, CEP164, in side view (*Figure 2D*). When top (or axial) view of the mother centriole was visualized, LAP-CEP15 formed a ring-like structure that is slightly smaller than the CEP164 ring (*Figure 2E*), which mirrors the ninefold symmetrical structure of the distal appendage (*Paintrand et al., 1992*). Similarly, NCS1 localized slightly distal to CEP164 (*Figure 2F*) as well as the binding partner, CEP89 (*Figure 2G*), and slightly proximal to the centriole-associated vesicle marker, RAB34 (*Stuck et al., 2021*; *Kanie et al., 2025*; *Figure 2H*). Like CEP15, NCS1 formed a slightly smaller ring than CEP164 (*Figure 2I*). Consistent with this, the ring diameter of NCS1 and CEP15 was 319.5 ± 7.7 nm (*n* = 13, average ± SEM) and 348.8 ± 8.0 nm (*n* = 16, average ± SEM), respectively (see Figure 1C of *Kanie et al., 2025*). It is notable that both CEP15 and NCS1 also localized to the region close to subdistal appendage in some but not all centrioles (*Figure 2D, F*), consistent with what was observed for CEP89 localization (*Chong et al., 2020*). This near subdistal appendage localization explains why CEP15 was previously classified as a subdistal appendage protein (*Gupta et al., 2015*). These results suggest that NCS1 localizes to the distal appendage and more precisely to a position sandwiched between CEP89 and the RAB34 positive vesicle (*Figure 2J*).

We next determined the hierarchy of the three proteins. Centriolar localization of CEP89 was not affected by depletion of either NCS1 nor CEP15 (*Figure 2K* and *Figure 2—figure supplement 2A*). NCS1 failed to localize to the mother centriole without altering its cytoplasmic localization in *CEP89* knockout cells but not in *CEP15* knockout cells (*Figure 2L* and *Figure 2—figure supplement 2B*), indicating that CEP89 recruits NCS1 to the distal appendage. The lack of NCS1 localization at the centriole in *CEP89* knockouts cells was rescued by expressing untagged CEP89 (*Figure 2L* and *Figure 2—figure supplement 2B*). CEP15 localization required CEP89, but not NCS1 (*Figure 2M* and *Figure 2—figure supplement 2C*). This is further corroborated by the structural modeling using Alphafold2, which shows that CEP89 interacts with both NCS1 and CEP15 whereas CEP15 does not interact with NCS1 (*Figure 2O*; *Figure 2—figure supplement 3*). The expression level of neither NCS1 nor CEP15 was affected by CEP89 depletion (*Figure 2—figure supplement 2D, E*). These results suggest that both NCS1 and CEP15 are recruited to the distal appendage by CEP89 (*Figure 2N*). We also tested whether the three proteins affect localization of other distal appendage proteins and found that the localization of other distal appendage proteins were unchanged in cells deficient in CEP89, NCS1, or CEP15 (*Figure 2—figure supplement 4A–C*) (see also Figure 2A–L of *Kanie et al., 2025*). Similar to the centriolar localization of CEP89, which was significantly reduced in *CEP83* or *SCLT1* knockouts, NCS1 localization was also greatly diminished in these knockouts (*Figure 2—figure supplement 4D*). This is consistent with the observation that CEP83-SCLT1 module serves as a structural component of the distal appendage (see *Kanie et al., 2025* for the detail). In contrast, the feedback complex CEP164-TTBK2 (see *Kanie et al., 2025* for the detail) was required for proper centriolar localization of NCS1 (*Figure 2—figure supplement 4D*) but not for CEP89 (Figure 2F of *Kanie et al., 2025*). Given that CEP89 is a substrate of TTBK2 (*Bernatik et al., 2020*; *Lo et al., 2019*), this might suggest that phosphorylation of TTBK2 could affect the interaction between CEP89 and NCS1. It is also possible that NCS1 may be a phosphorylation target of TTBK2. These questions warrant future investigation.

## NCS1 is important for efficient preciliary vesicle recruitment at the distal appendage

We next sought to understand the role of NCS1 at the distal appendage and performed kinetic analysis of ciliation in control (sgGFP) and knockouts of *CEP89*, *NCS1*, or *CEP15* in RPE cells, in which

serum starvation induces cilium formation (*Figure 3A*). Control RPE cells form cilia over 24 hr after serum starvation, and almost of all cells completed cilium formation between 24 and 48 hr. *CEP89* and *NCS1* knockouts displayed a notable delay in initiating ciliation (see 12 hr in *Figure 3A*), but gradually catch up on ciliogenesis and exhibited only mild ciliary formation defects at later time points (see 48 hr in *Figure 3A*). The cilium formation defect was rescued by expressing untagged CEP89 in *CEP89* knockouts or untagged NCS1 in *NCS1* knockouts (*Figure 3B*). This kinetic defect is strongly consistent with the knockouts of several other distal appendage proteins, namely ANKRD26 and FBF1 (Figure 5A, B of *Kanie et al., 2025*). The ciliary length of *CEP89*, *NCS1*, or *CEP15* knockout cells was comparable to that of control cells (*Figure 3—figure supplement 1A*). Consistent with the cells deficient in ANKRD26 or FBF1 (see Figure 5—figure supplement 1 of *Kanie et al., 2025*), ciliary ARL13B signal intensity was reduced in *CEP89* or *NCS1* knockouts even after the cells complete cilium formation (*Figure 3—figure supplement 1B–D*). This suggests that even though the *NCS1* or *CEP89* knockouts can eventually form primary cilia, those cilia may be functionally different from wild-type cilia. *CEP15* knockouts showed similar but much milder kinetic defect of cilium formation than *CEP89* or *NCS1* knockouts, therefore, we focused on NCS1 in the subsequent investigation.

We then sought to understand how NCS1 is involved in cilium formation. In an accompanying paper (*Kanie et al., 2025*), we showed that CEP89 participates in cilium formation by regulating preciliary vesicle recruitment without affecting IFT88::CEP19 recruitment, important steps that require distal appendage proteins (*Schmidt et al., 2012*; *Dateyama et al., 2019*) (see Figure 5 of *Kanie et al., 2025*). We tested if NCS1 has similar roles to its binding partner, CEP89. Indeed, *NCS1* knockouts exhibited moderate preciliary vesicle recruitment defect, similar to *CEP89* knockouts, when assessed using RAB34 as the vesicle marker (*Figure 3C*). Transmission electron microscopy analysis confirmed the vesicle recruitment defect in *NCS1* knockouts (*Figure 3D, E*). The presence of fused vesicles, albeit much lower percentage than control cells, in *NCS1* knockouts (*Figure 3D, E*) suggests that NCS1 is important for recruitment but not fusion of the vesicle. Removal of CP110, which is believed to act as a cap of axonemal microtubule, was partially, but measurably affected in *NCS1* knockout cells (*Figure 3F*), consistent with the fact that CP110 removal is in part downstream of the vesicle recruitment (Figure 4D of *Kanie et al., 2025*). Neither IFT88 nor CEP19 recruitment to the centriole was affected in *NCS1* knockout cells (*Figure 3G, H*). These results suggest that NCS1 plays an important role in cilium formation by regulating preciliary vesicle recruitment, but not other known processes of cilium formation, consistent with the role of CEP89 in preciliary vesicle recruitment.

## Yet unknown distal appendage proteins may compensate the lack of NCS1 in preciliary vesicle recruitment

As we showed in an accompanying paper, the distal appendage is indispensable for the recruitment of the preciliary vesicle to the mother centriole. Virtually no RAB34-positive ciliary vesicle was observed at mother centriole in cells deficient in CEP164, CEP83, or TTBK2, which are critical for structural integrity of the distal appendage (Figure 5C of *Kanie et al., 2025*). This phenotype is much stronger than what we observed in *CEP89* or *NCS1* knockouts (*Figure 3C*), suggesting that some other distal appendage proteins may compensate the lack of NCS1 for the preciliary vesicle recruitment. To address this question, we created cells lacking both NCS1 and each of the other distal appendage proteins (FBF1, CEP89, ANKRD26, KIZ, LRRC45) as well as the distal appendage associated protein, INPP5E (*Figure 4*). We omitted the integral components of the distal appendage proteins (CEP164, TTBK2, CEP83, SCLT1) from the analysis, as the knockouts of these proteins showed very strong preciliary vesicle recruitment defects on their own, therefore making it difficult to test if the NCS1 depletion shows additive effects. Cilium formation assay revealed that depletion of NCS1 decreased the percentage of ciliated cells in each single knockout cells, except *CEP89* (*Figure 4B, C*). This suggests that CEP89, but no other distal appendage protein regulates the same ciliary formation pathway as NCS1. Similarly, depletion of NCS1 decreased the RAB34-positive centriole in each of the single distal appendage knockout cells, except *CEP89* knockouts (*Figure 4D*). These results suggest that yet unknown distal appendage proteins would be required to compensate for the preciliary vesicle recruitment defect of *NCS1* knockout cells. Another possibility is that one or more integral components (CEP164, TTBK2, SCLT1, and CEP83) may be directly involved in preciliary vesicle recruitment. These hypotheses warrant future investigation.

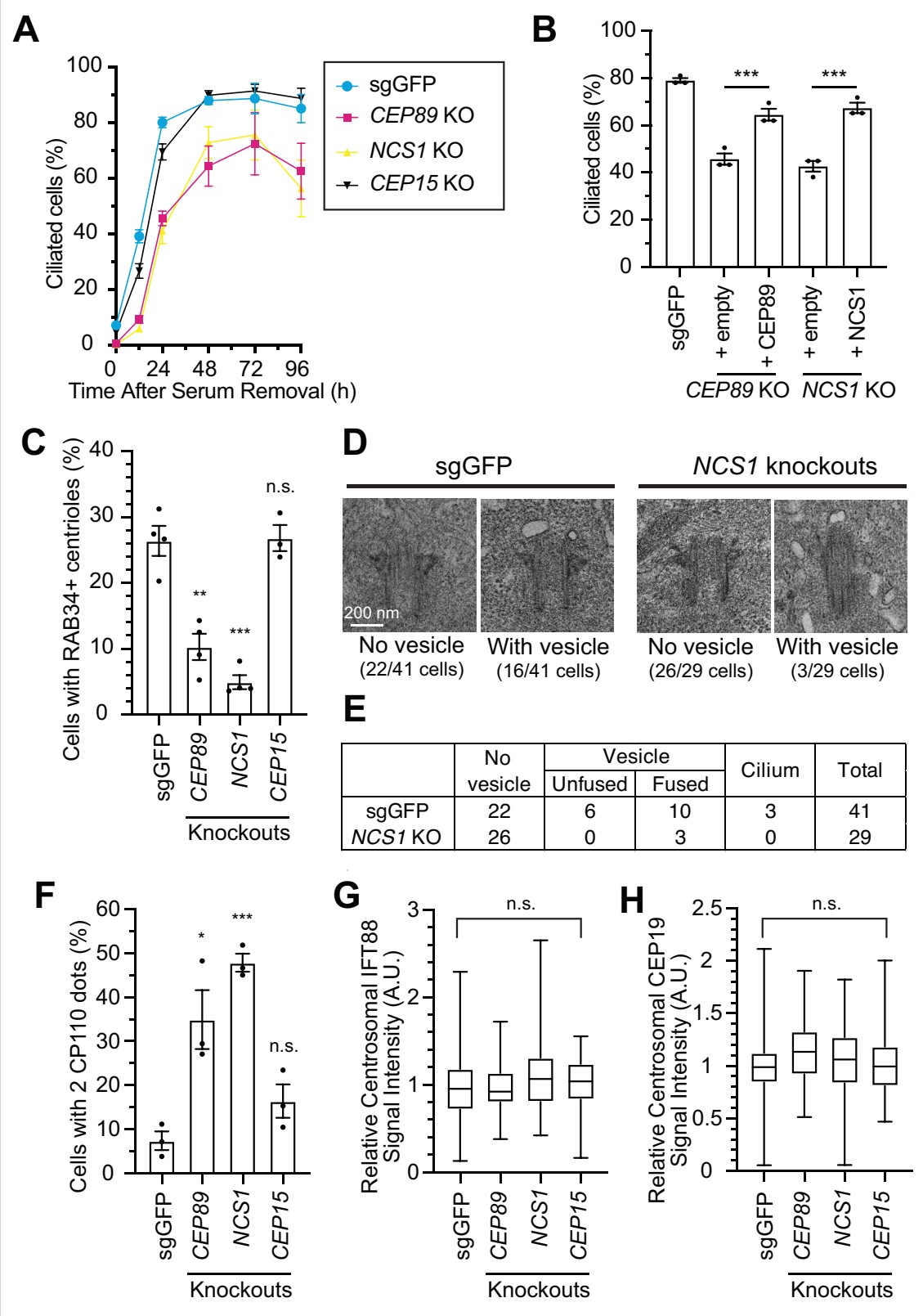

**Figure 3.** NCS1 is important for preciliary vesicle recruitment, but not for IFT88/CEP19 recruitment. (**A**) Time course of cilium formation assay in control (sgGFP) and indicated knockout retinal pigment epithelia (RPE) cells. The indicated cells were serum starved for 12, 24,48,72,96 hr, fixed, stained with α-ARL13B (to mark cilium) and α-CEP170 (to mark centriole), and imaged via wide-field microscopy. Data averaged from four independent experiments. Error bars represent ± SEM. Statistics obtained through comparing between each knockout and control by Welch's *t*-test. The raw data, experimental

*Figure 3 continued on next page*

*Figure 3 continued*

conditions, and detailed statistics are available in *Figure 3—source data 1*. (**B**) Cilium formation assay in control (sgGFP) and indicated knockout RPE cells serum starved for 24 hr. Data averaged from three independent experiments, and each black dot indicates the value from the individual experiment. Error bars represent ± SEM. Statistics obtained by Welch's *t*-test. The raw data, experimental conditions, and detailed statistics are available in *Figure 3—source data 2*. (**C**) Preciliary vesicle recruitment assay in control (sgGFP) or indicated knockout RPE cells grown to confluence (without serum starvation). The data are averaged from four independent experiments, and each black dot indicates the value from each individual experiment. Error bars represent ± SEM. Statistics obtained through comparing between each knockout and control by Welch's *t*-test. The raw data, experimental conditions, and detailed statistics are available in *Figure 3—source data 3*. (**D**) Transmission electron microscopy analysis of the mother centriole in control (sgGFP) or *NCS1* knockout RPE cells serum starved for 3 hr. The representative images of the mother centrioles without (left) or with (right) vesicles at the distal appendage are shown. Scale: 200 nm. (**E**) Quantification of the data from the experiments shown in panel D. The raw data and detailed statistics are available in *Figure 3—source data 4*. This experiment was synchronized with the experiment shown in Figure 4C of *Kanie et al., 2025*, hence the values for sgGFP are exactly the same as the ones shown in *Kanie et al., 2025*. (**F**) CP110 removal assay in control (sgGFP) and indicated knockout RPE cells serum starved for 24 hr. Data are averaged from three independent experiments, and each black dot indicates the value from the individual experiment. Error bars represent ± SEM. Statistics obtained through comparing between each knockout and control by Welch's *t*-test. The raw data, experimental conditions, and detailed statistics are available in *Figure 3—source data 5*. Quantification of the centrosomal signal intensity of IFT88 (**G**) or CEP19 (**H**) in control (sgGFP) and indicated knockout RPE cells serum starved for 24 hr. The data are combined from three independent experiments. Statistical significance was calculated from nested *t*-test. The raw data, experimental conditions, and detailed statistics are available in *Figure 3—source data 6* and *Figure 3—source data 7*. A.U., arbitrary units; n.s., not significant; *p < 0.05, **p < 0.01, ***p < 0.001.

The online version of this article includes the following source data and figure supplement(s) for figure 3:

**Source data 1.** Raw quantification data, immunofluorescence conditions, and detailed statistics of the experiment shown in *Figure 3A*.

**Source data 2.** Raw quantification data, immunofluorescence conditions, and detailed statistics of the experiment shown in *Figure 3B*.

**Source data 3.** Raw quantification data, immunofluorescence conditions, and detailed statistics of the experiment shown in *Figure 3C*.

**Source data 4.** Raw quantification data and detailed statistics of the experiment shown in *Figure 3E*.

**Source data 5.** Raw quantification data, immunofluorescence conditions, and detailed statistics of the experiment shown in *Figure 3F*.

**Source data 6.** Raw quantification data, immunofluorescence conditions, and detailed statistics of the experiment shown in *Figure 3G*.

**Source data 7.** Raw quantification data, immunofluorescence conditions, and detailed statistics of the experiment shown in *Figure 3H*.

**Figure supplement 1.** Quantification of ciliary signal intensity of ARL13B in CEP89 and *NCS1* knockouts.

**Figure supplement 1—source data 1.** Immunofluorescence conditions, raw quantification data, and detailed statistics of the experiment shown in *Figure 3—figure supplement 1A*.

**Figure supplement 1—source data 2.** Immunofluorescence conditions, raw quantification data, and detailed statistics of the experiment shown in *Figure 3—figure supplement 1B–D*.

## NCS1 captures preciliary vesicle via its myristoylation motif

Since NCS1 is myristoylated, we wondered whether the membrane association motif is necessary for NCS1 to recruit the preciliary vesicle. We tested this hypothesis by creating a myristoylation defective mutant (NCS1-G2A). We tested whether the mutant indeed lost the ability to bind membrane using differential centrifugation following nitrogen cavitation. We first determined which fraction is the most optimal to assess membrane association in our experimental setting. While the microsomal fraction prepared from the pellet following ultracentrifugation at 100,000 × *g* is often used to analyze membrane fraction of the cells (*Graham, 2015*), the plasma membrane marker, Epidermal Growth Factor Receptor (EGFR), was enriched mostly in the pellet following centrifugation at 15,000 × *g* in our experiment (*Figure 5A*; see Methods for further explanation of the technical design). In addition, NCS1-G2A was fractionated in the 100,000 × *g* pellet to similar extent as wild-type NCS1 (*Figure 5A*), indicating that centrifugation at 100,000 × *g* may also pellet some soluble proteins, even though cytoplasmic protein RabGDI remained in the 100,000 × *g* supernatant. Thus, we decided to use 15,000 × *g* pellet to assess membrane fraction in our experiment. In control cells (sgSafe), both NCS1 and CEP89 were found in both soluble fraction (15,000 × *g* supernatant) and membrane fraction (15,000 × *g* pellet) (*Figure 5A*). In contrast, NCS1-G2A was only found in the soluble fraction (15,000 × *g* supernatant) (*Figure 5A*), suggesting that myristoylation is required for membrane localization of NCS1. Interestingly, CEP89 was only found in the soluble fraction (15,000 × *g* supernatant) in *NCS1* knockout cells expressing either empty vector or NCS1-G2A, but not NCS1-WT. This suggests that membrane localization of CEP89 requires NCS1 with an intact myristoylation motif. When expressed at similar level to endogenous NCS1 (*Figure 5B*), both wild-type and the myristoylation defective (G2A) NCS1 localizes to the mother centriole to a similar extent (*Figure 5C, D*). However, the myristoylation defective

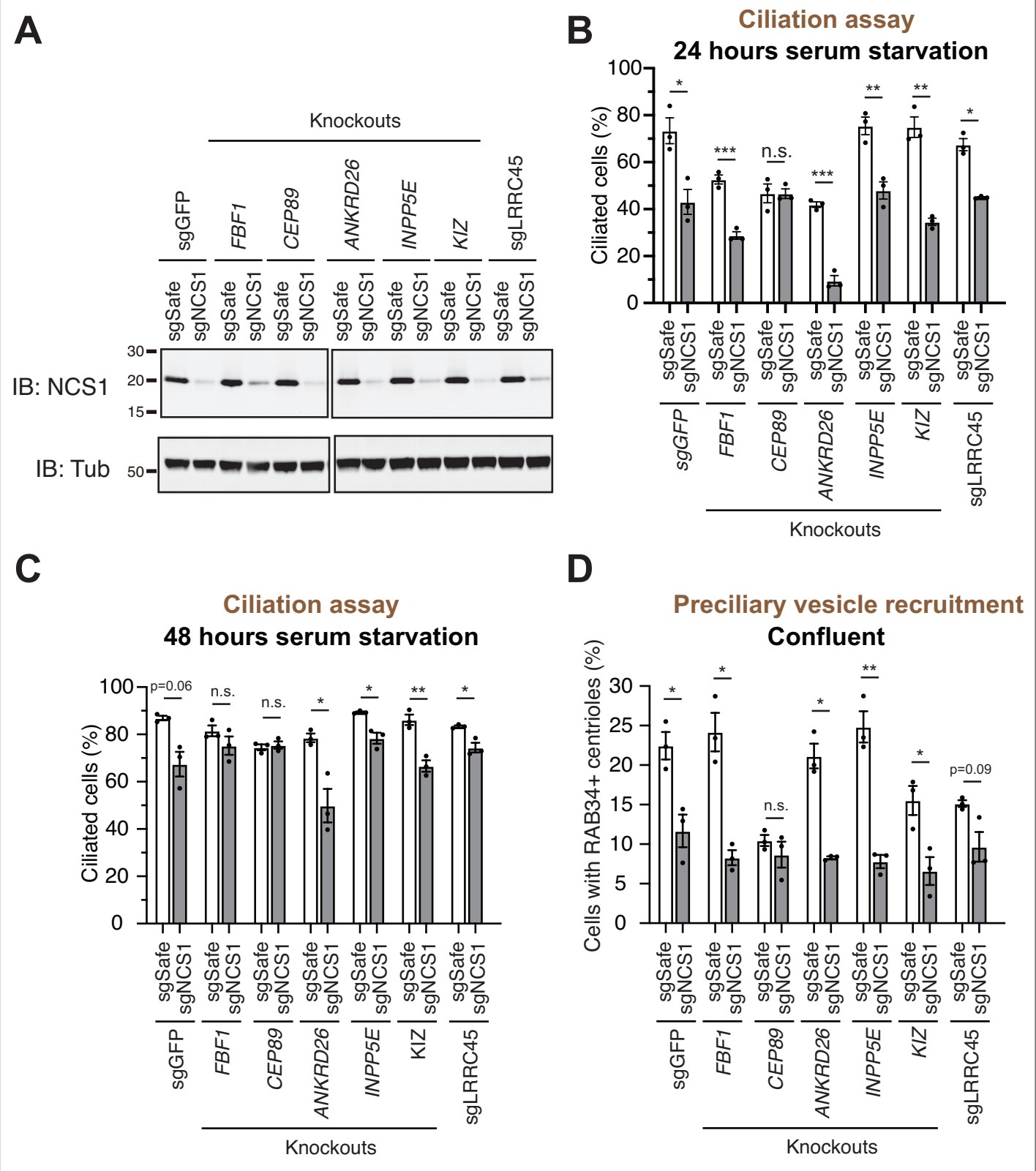

**Figure 4.** A preciliary vesicle recruitment defect in *NCS1* knockout cells is compensated by yet unknown distal appendage proteins. (**A**) Immunoblot (IB) analysis of expression of NCS1 (IB: NCS1) and α-tubulin (IB: Tub) in control (sgGFP) or indicated knockout retinal pigment epithelia (RPE) cells stably expressing either sgSafe (non-targeting) or sgNCS1. The cells were grown to confluence (without serum starvation), lysed and analyzed by immunoblot. Molecular weights (kDa) estimated from a protein marker are indicated. The raw unedited blots can be found in *Figure 4—source data 1* and *Figure*

*Figure 4 continued on next page*

*Figure 4 continued*

**4—source data 2**. (**B, C**) Cilium formation assay in control (sgGFP) and indicated knockout RPE cells stably expressing either sgSafe (non-targeting) or sgNCS1. The cells were serum starved for 24 (**B**) or 48 (**C**) hr. Data averaged from three independent experiments, and each black dot indicates the value from the individual experiment. Error bars represent ± SEM. Statistics obtained by Welch's *t*-test. The raw data, experimental conditions, and detailed statistics are available in *Figure 4—source data 3* and *Figure 4—source data 4*. (**D**) Preciliary vesicle recruitment assay in indicated knockout RPE cells stably expressing either sgSafe (control) or sgNCS1. Cells were grown to confluence (without serum starvation). Data are averaged from three independent experiments. Error bars represent ± SEM. Statistics obtained by Welch's *t*-test. The raw data, experimental conditions, and detailed statistics are available in *Figure 4—source data 5*. n.s., not significant; *p < 0.05, **p < 0.01, ***p < 0.001.

The online version of this article includes the following source data for figure 4:

**Source data 1.** The original files of the full raw unedited blots shown in *Figure 4A*.

**Source data 2.** The uncropped blots with boxes that indicate the regions displayed in *Figure 4A*.

**Source data 3.** Raw quantification data, immunofluorescence conditions, and detailed statistics of the experiment shown in *Figure 4B*.

**Source data 4.** Raw quantification data, immunofluorescence conditions, and detailed statistics of the experiment shown in *Figure 4C*.

**Source data 5.** Raw quantification data, immunofluorescence conditions, and detailed statistics of the experiment shown in *Figure 4D*.

mutant almost completely failed to rescue ciliation and preciliary vesicle recruitment defect of *NCS1* knockout cells (*Figure 5E, F*). These data suggest that NCS1 recruits preciliary vesicle to the distal appendage via its myristoylation motif. Structural modeling of the NCS1–CEP89 complex reveals a distinct functional organization of NCS1 (*Figure 5G*). The three intact $Ca^{2+}$-binding EF hands and the Gly2 myristoylation site cluster on one face of NCS1, while the CEP89-binding pocket occupies the opposite face. This arrangement suggests that NCS1 can simultaneously engage in membrane association and calcium sensing while interacting with CEP89. This architecture provides a molecular basis for understanding how the myristoylation-dependent membrane association of NCS1 can facilitate the recruitment of preciliary vesicles at the distal appendage, as supported by our biochemical and cellular analyses.

## Calcium binding is needed for stability of NCS1

In addition to myristoylation motif, human NCS1 also has three functional and one apparently non-functional (due to the mutation in critical amino acids needed for co-ordination bond with calcium) EF-hand motifs (*Bourne et al., 2001*). Several other NCS family proteins, including Recoverin and Hippocalcin, show a calcium-myristoyl switch (*Ames et al., 1997*; *O'Callaghan et al., 2003*). In this mechanism, the sequestered myristoylation motif is exposed to allow the protein to bind membrane upon calcium binding. It has been proposed that NCS1 may employ a similar molecular switch. While the structure of fission yeast Ncs1, solved by nuclear magnetic resonance spectroscopy showed the calcium-myristoyl switch (*Lim et al., 2011*), several lines of evidence suggest the absence of that type of switch in budding yeast and mammalian NCS1 (*Ames et al., 2000*; *O'Callaghan et al., 2002*; *Lemire et al., 2016*). Since a myristoylation defective mutant of NCS1 failed to form cilia efficiently without affecting its centrosomal localization (*Figure 5C, E*), we tested if there is a calcium-myristoylation switch by making various EF-hand mutations, where each or combination of the three active EF-hand motif was disabled by mutating invariant glutamate at -z position to glutamine (E84Q for the first, E120Q for the second, and E168Q for the third active EF-hand mutation). We expressed wild-type or EF-hand mutants of untagged NCS1 in *NCS1* knockout cells and detected the expression and the localization of each mutant using α-NCS1 antibody. The wild-type and the mutants of NCS1 were functionally tested via ciliation assay rather than the preciliary vesicle recruitment assay, because the ciliation assay is much less variable than the vesicle recruitment assay. The E84Q mutant as well as any double and triple EF-hand mutants were highly destabilized (see input in *Figure 5—figure supplement 1A*), and the centriolar NCS1 signal intensity of the mutants was reduced in parallel to their diminished expression level (*Figure 5—figure supplement 1B*). This suggests that the first EF hand is indispensable for stability of NCS1. E120Q mutant had similar expression level, but its centrosomal signal was significantly reduced (*Figure 5—figure supplement 1B*), consistent with its diminished interaction with CEP89 (see IP: GFP in *Figure 5—figure supplement 1A*), a protein that recruits NCS1 to the mother centriole (*Figure 2L*). This suggests that the second EF hand is involved in keeping its structure to interact with CEP89. E168Q had a negligible effect in stability and localization of NCS1. The ciliation assay revealed that none of single EF-hand mutants showed significant cilium formation

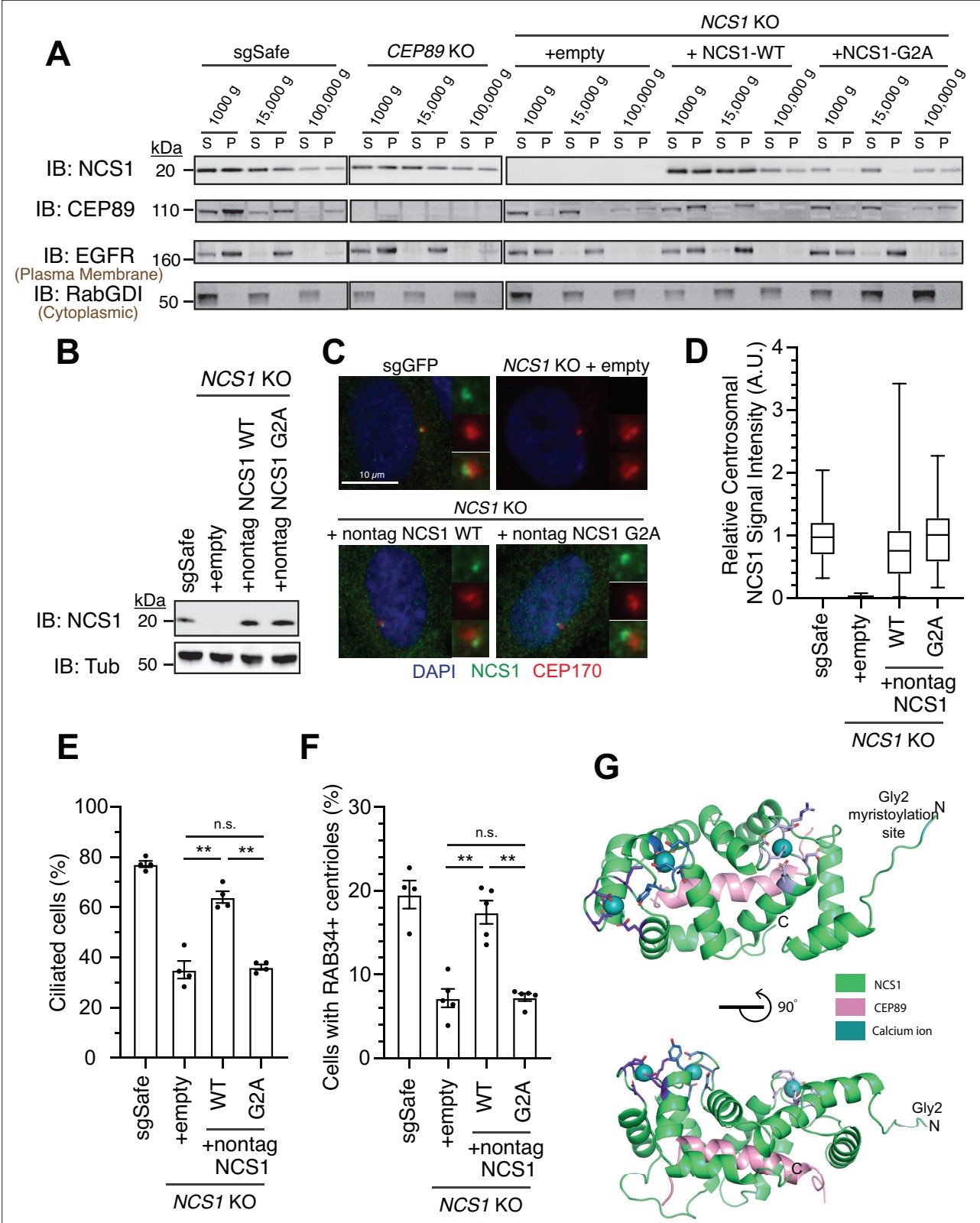

**Figure 5.** NCS1 captures the preciliary vesicle via its myristoylation motif. (**A**) Immunoblot (IB) analysis of expression of NCS1, CEP89, EGFR, and RabGDI. The control (sgSafe) and indicated knockout retinal pigment epithelia (RPE) cells were grown to confluence, lysed by nitrogen cavitation, and fractionated by differential centrifugation at 1000, 15,000, and 100,000 × *g*. S: supernatant; P: pellet. Molecular weights (kDa) estimated from a protein marker are indicated. EGFR and RabGDI serve as representative markers for plasma membrane or cytoplasmic proteins, respectively. The raw unedited

*Figure 5 continued on next page*

*Figure 5 continued*

blots can be found in *Figure 5—source data 1* and *Figure 5—source data 2*. (**B**) Immunoblot (IB) analysis of expression of NCS1 and α-tubulin in control (sgSafe) or indicated RPE cells. Molecular weights (kDa) estimated from a protein marker are indicated. The raw unedited blots can be found in *Figure 5—source data 3* and *Figure 5—source data 4*. (**C**) Immunofluorescence images taken via wide-filed microscopy in the cells described in (**B**) serum starved for 24 hr. Insets at the right panels are the enlarged images of the mother centriole. Scale bar: 10 µm. (**D**) Box plots showing centrosomal signal intensity of NCS1 in cells described in (**B**) that were serum starved for 24 hr. A.U., arbitrary units. The data from a representative experiment are shown. The raw data and experimental conditions are available in *Figure 5—source data 6*. (**E**) Cilium formation assay in the cells described in (**B**) serum starved for 24 hr. Data averaged from four independent experiments, and each black dot indicates the value from the individual experiment. Error bars represent ± SEM. Statistics obtained by Welch's *t*-test. The raw data, experimental conditions, and detailed statistics are available in *Figure 5—source data 7*. (**F**) Preciliary vesicle recruitment assay in the cells described in (**B**) grown to confluence (without serum starvation). Data averaged from five independent experiments. Error bars represent ± SEM. Statistics obtained through comparing between each knockout and control by Welch's *t*-test. The raw data, experimental conditions, and detailed statistics are available in *Figure 5—source data 8*. (**G**) Cartoon representation of two perpendicular views of the CEP89-NCS1structural model with the myristoylation site (glycine 2) highlighted in cyan. Calcium ions are shown as spheres and EF-hand motifs of NCS1 are shown as sticks and colored in different shades of blue. The NCS1-binding helix of CEP89 is shown in pink. n.s., not significant; **p < 0.01.

The online version of this article includes the following source data and figure supplement(s) for figure 5:

**Source data 1.** The original files of the full raw unedited blots shown in *Figure 5A*.

**Source data 2.** The uncropped blots with boxes that indicate the regions displayed in *Figure 5A*.

**Source data 3.** The original files of the full raw unedited blots shown in *Figure 5B*.

**Source data 4.** The uncropped blots with boxes that indicate the regions displayed in *Figure 5B*.

**Source data 5.** Immunofluorescence conditions in the experiment shown in *Figure 5C*.

**Source data 6.** Raw quantification data and immunofluorescence conditions of the experiment shown in *Figure 5D*.

**Source data 7.** Raw quantification data, immunofluorescence conditions, and detailed statistics of the experiment shown in *Figure 5E*.

**Source data 8.** Raw quantification data, immunofluorescence conditions, and detailed statistics of the experiment shown in *Figure 5F*.

**Figure supplement 1.** Calcium is required mainly for the stability of NCS1.

**Figure supplement 1—source data 1.** The original files of the full raw unedited blots shown in *Figure 5—figure supplement 1A*.

**Figure supplement 1—source data 2.** The uncropped blots with boxes that indicate the regions displayed in *Figure 5—figure supplement 1A*.

**Figure supplement 1—source data 3.** Immunofluorescence conditions and raw quantification data of the experiment shown in *Figure 5—figure supplement 1B*.

**Figure supplement 1—source data 4.** Immunofluorescence conditions, raw quantification data, and detailed statistics of the experiment shown in *Figure 5—figure supplement 1C*.

defect (*Figure 5—figure supplement 1C*) despite the partial reduction of centriolar signal intensity for E84Q and E120Q. The double or triple mutants almost completely failed to rescue ciliation defect of *NCS1* knockout cells (*Figure 5—figure supplement 1C*), reflecting their very low expression level (*Figure 5—figure supplement 1A*). These results suggest that calcium binding is primarily required for the structure and stability of NCS1 and NCS1 does not clearly exhibit a calcium-myristoylation switch. The structural role of calcium on NCS1 is largely consistent with NCS1's high binding affinity to calcium (−90 nM) (*Aravind et al., 2008*).

## NCS1 is recruited to the centriole in a microtubule-independent manner where it captures preciliary vesicles

We next sought to understand where NCS1 captures preciliary vesicle. One possibility is that NCS1 captures the vesicle in the cytoplasm and then traffics it to the centriole by dynein-dependent transport via microtubules. Another possibility is that NCS1 traffics to the centriole first and then captures preciliary vesicles. To distinguish these two possibilities, we treated RPE cells with nocodazole to destabilize microtubules, as microtubules were previously shown to be indispensable for preciliary vesicle recruitment (*Wu et al., 2018*). Consistent with the previous report (*Wu et al., 2018*), destabilization of microtubule by nocodazole (*Figure 6A, B*) immediately inhibited preciliary vesicle recruitment and subsequent cilium formation (*Figure 6C, D*), suggesting that preciliary vesicles are trafficked to the mother centriole via microtubules. In contrast, centriolar NCS1 signal was gradually increased upon serum starvation even in the presence of nocodazole (*Figure 6E*), suggesting that NCS1 accumulates at the distal appendage by microtubule-independent mechanisms, possibly by

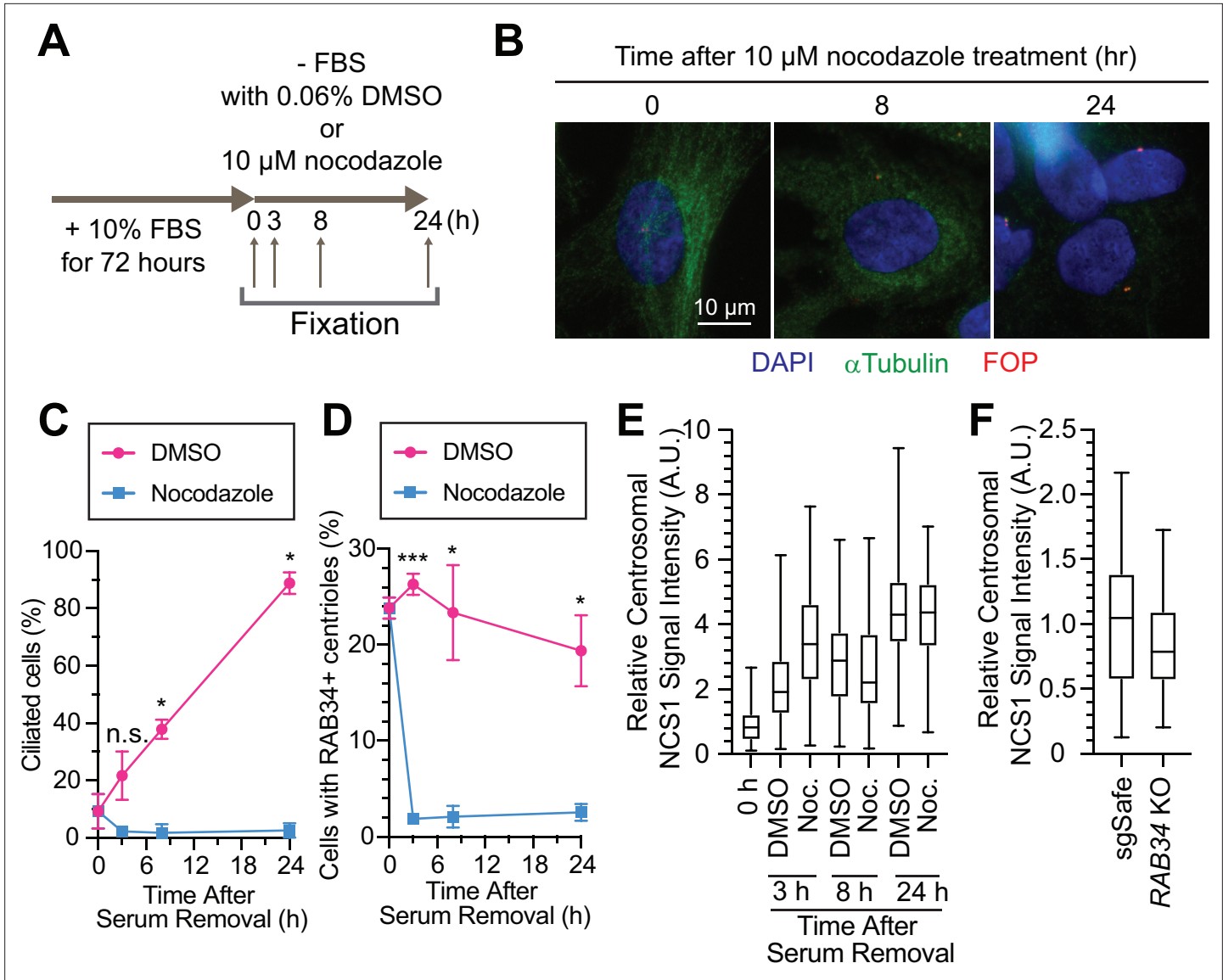

**Figure 6.** Preciliary vesicle, but not NCS1 and CEP89, is recruited to the centriole via microtubules. (**A**) A cartoon showing the method used to test the requirement of microtubules in preciliary vesicle recruitment and NCS1 localization. Retinal pigment epithelia (RPE) cells were cultured in media containing 10% fetal bovine serum (FBS) for 72 hr, then serum starved (−FBS) for indicated times in the presence of nocodazole or dimethyl sulfoxide (DMSO). (**B**) Immunofluorescence images taken via wide-filed microscopy. RPE cells were cultured as shown in (**A**), fixed, and stained with antibodies against α-tubulin and FGFR1OP (FOP). Scale bar: 10 μm. (**C**) The time course of cilium formation in cells treated with either DMSO (magenta) or nocodazole (blue). The cells were fixed at indicated time points, stained with α-ARL13B (to mark cilium) and α-CEP170 (to mark centriole), and imaged via wide-field microscopy. Data averaged from three independent experiments. Error bars represent ± SEM. Statistics obtained through comparing between DMSO and nocodazole treated cells at each time point by Welch's *t*-test. The raw data, experimental conditions, and detailed statistics are available in *Figure 6—source data 2*. (**D**) The time course of preciliary vesicle recruitment in cells treated with either DMSO (magenta) or nocodazole (blue). The cells were fixed at indicated time points, stained with α-RAB34 (to mark the centriole-associated vesicle) and α-CEP170 (to mark centriole), and imaged via wide-field microscopy. Data are averaged from three independent experiments. Error bars represent ± SEM. Statistics obtained through comparing between DMSO and nocodazole treated cells at each time point by Welch's *t*-test. The raw data, experimental conditions, and detailed statistics are available in *Figure 6—source data 3*. (**E**) Box plots showing centrosomal signal intensity of NCS1 in RPE cells prepared using the method described in (**A**). The data from the representative experiment are shown. The raw data and experimental conditions are available in *Figure 6—source data 4*. (**F**) Quantification of the centrosomal signal intensity of NCS1 in control or *RAB34* knockout RPE cells serum starved for 24 hr. The data from the representative experiment are shown. The raw data and experimental conditions are available in *Figure 6—source data 5*. A.U., arbitrary units; n.s., not significant; *p < 0.05, ***p < 0.001.

The online version of this article includes the following source data for figure 6:

**Source data 1.** Immunofluorescence conditions in the experiment shown in *Figure 6B*.

*Figure 6 continued on next page*

*Figure 6 continued*

**Source data 2.** Raw quantification data, immunofluorescence conditions, and detailed statistics of the experiment shown in *Figure 6C*.

**Source data 3.** Raw quantification data, immunofluorescence conditions, and detailed statistics of the experiment shown in *Figure 6D*.

**Source data 4.** Raw quantification data and immunofluorescence conditions of the experiment shown in *Figure 6E*.

**Source data 5.** Raw quantification data and immunofluorescence conditions of the experiment shown in *Figure 6F*.

diffusion, similar to the previously proposed diffusion-to-capture model of IFT trafficking to the ciliary base (*Hibbard et al., 2021*). Because NCS1 is an N-terminally myristoylated protein, we also considered whether NCS1 might be trafficked to the distal appendage through UNC119, a chaperone that binds to N-myristoylated ciliary proteins like NPHP3 and cystin and traffics them to the primary cilium (*Wright et al., 2011*). We conclude that NCS1 is unlikely to use the UNC119 pathway, as we did not observe UNC119A/UNC119B in our AP/MS analysis of CEP89 (*Figure 1—source data 2*), nor in purifications of UNC119A/UNC119B proteins themselves (*Wright et al., 2011*). In a complementary approach to the microtubule destabilization, we tested if NCS1 localizes to the distal appendage even if the preciliary vesicle recruitment is inhibited by RAB34 depletion (see Figure 4B, C of *Kanie et al., 2025*). We confirmed that centriolar NCS1 was comparable between control (sgSafe) and *RAB34* knockout cells (*Figure 6F*), suggesting that NCS1 is recruited to the centriole independently from the ciliary vesicle. These results suggest that NCS1 moves to the distal appendage possibly by diffusion or an alternative trafficking mechanism and there it captures the preciliary vesicle that is trafficked to the centriole by microtubule-dependent trafficking.

## NCS1 localizes to the ciliary base in neuronal and non-neuronal cells

While the majority of papers to date focused on the role of NCS1 in neurons given its original discovery as a protein that facilitates neurotransmitter release (*Pongs et al., 1993*), expression analysis revealed that the protein is expressed ubiquitously in non-neuronal tissues (*Gierke et al., 2004*). Since we discovered that NCS1 localizes to the centriole, a major microtubule organizing center in animal cells (*Bornens, 2012*), we tested the expression and the localization of NCS1 in neuronal and non-neuronal cell types. We first tested the expression of NCS1 in various murine tissues and confirmed that NCS1 is expressed in a wide range of tissues (*Figure 7—figure supplement 1A*). The low expression level of NCS1 in liver, skeletal muscle, and fat might reflect that most of the cells (hepatocyte, myocyte, and adipocytes, respectively) in those tissues do not retain cilia, or because of the difference in ratio between intra- and extracellular proteins. When we performed immunofluorescence (IF) assay, NCS1 localized to both cytoplasm and the ciliary base (a dot next to ciliary markers, ARL13B or AC3) in isolated hippocampal neuron (top panel in *Figure 7A*) as well as cells in hypothalamus (*Figure 7B*) and the dentate gyrus of the hippocampus (*Figure 7—figure supplement 2C*). Both cytoplasmic and the centriole signal was specific for NCS1 as we detected no signal in *NCS1* knockout cells (bottom panel in *Figure 7A, B* and *Figure 7—figure supplement 2C*). NCS1 also localized to the ciliary base in virtually all the non-neuronal ciliated cells that we tested, including kidney epithelia, pancreatic islet cells, airway epithelia, ependymal cells, and mouse embryonic fibroblasts (MEFs) (*Figure 7C, D*, *Figure 7—figure supplement 2A*). Only ciliated cells where we failed to detect NCS1 at the ciliary base were photoreceptor cells (*Figure 7—figure supplement 2E*). We next sought to determine whether NCS1 is involved in cilium formation in those cells, as we saw in RPE cells (*Figure 3A, B*). In MEFs, cilium formation is modestly perturbed in Ncs1-depleted cells (*Figure 7E*), consistent with the kinetic cilium formation defect in RPE cells (*Figure 3A*). Decrease in ciliary ARL13B signal was also detected in $Ncs1^{-/-}$ MEFs (*Figure 7F*), similar to *NCS1* knockout RPE cells (*Figure 3—figure supplement 1B–D*). In contrast, we did not detect a measurable cilium formation defect in hippocampal neurons that lack Ncs1 (*Figure 7G*), whereas the signal of ciliary membrane protein, type III adenylyl cyclase (ADCY3) (*Bishop et al., 2007*; *Berbari et al., 2007*), was significantly decreased (*Figure 7H*). Other ciliary GPCRs, such as SSTR3 and GPR161, were also decreased in $Ncs1^{-/-}$ neurons, but did not show statistical significance with the small number of samples analyzed (*Figure 7—figure supplement 3A, B*). The difference in the cilium formation defect in different cell types (modest ciliation defects in RPE and MEFs, but no defect in neurons) as well as potential signaling function of NCS1 will be considered in the Discussion.

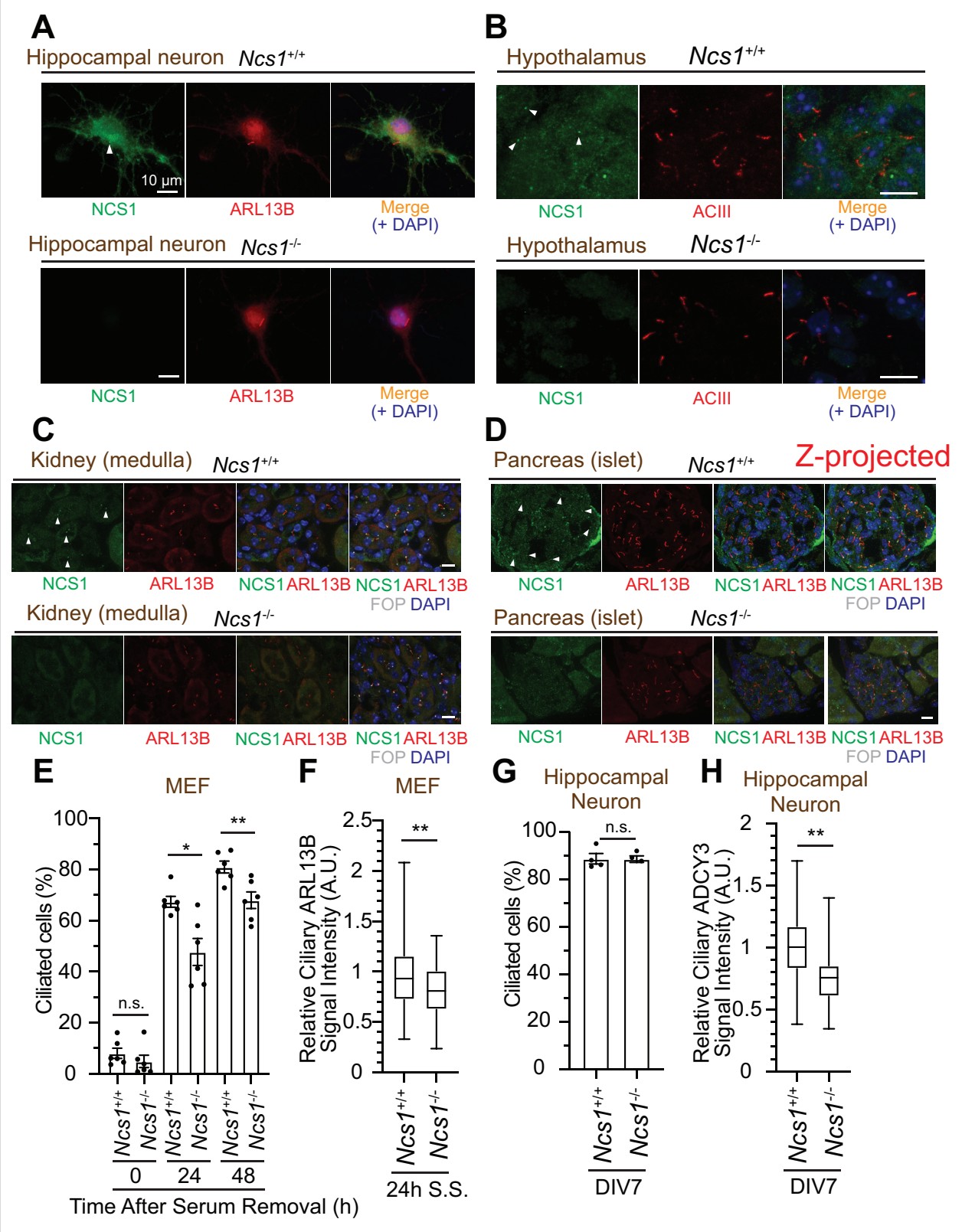

**Figure 7.** NCS1 localizes to the ciliary base in ciliated tissues and gets involved in cilium formation and ciliary membrane protein trafficking. (**A**) Immunofluorescence images of cultured hippocampal neurons taken by wide-field microscopy. The isolated hippocampal neurons from E18.5 *Ncs1*<sup>+/+</sup> or *Ncs1*<sup>−/−</sup> mice were fixed and stained for the indicated markers. Arrow indicates Ncs1 localization at the ciliary base. The individual image is from a representative z-slice. Scale bar: 10 µm. (**B–D**) Immunofluorescence images of indicated mouse tissues taken via spinning disk confocal

*Figure 7 continued on next page*

*Figure 7 continued*

microscopy. Tissue sections prepared from 8-week-old *Ncs1*$^{+/+}$ or *Ncs1*$^{-/-}$ mice with the method described in Materials and methods were stained for indicated markers. The images shown in (**D**) were created by maximum intensity z-projection. The other images were from representative z-slices. Arrowheads indicate NCS1 localization. Scale bar: 10 µm. (**E**) Cilium formation assay in *Ncs1*$^{+/+}$ or *Ncs1*$^{-/-}$ mouse embryonic fibroblasts (MEFs) serum starved for indicated time. Data averaged from six different MEFs per genotype. Each black dot indicates the value from the individual experiment. Error bars represent ± SEM. Statistics obtained through comparing between the two genotypes at each time point by Welch's *t*-test. The raw data, experimental conditions, and detailed statistics are available in *Figure 7—source data 5*. (**F**) Box plots showing ciliary signal intensity of ARL13B in *Ncs1*$^{+/+}$ or *Ncs1*$^{-/-}$ MEFs. The cells were serum starved for 24 hr, fixed, stained with α-ARL13B (to mark cilium) and α-CEP170 (to mark centriole), and imaged via wide-field microscopy. Data averaged from six different MEFs per genotype. Statistical significance was calculated from nested *t*-test. The raw data, experimental conditions, and detailed statistics are available in *Figure 7—source data 6*. (**G**) Cilium formation assay in isolated hippocampal neurons prepared from *Ncs1*$^{+/+}$ or *Ncs1*$^{-/-}$ E18.5 mouse embryos at 7 days in vitro (DIV). Data are averaged from four different hippocampal neurons per genotype. Each black dot indicates the value from the individual experiment. Error bars represent ± SEM. Statistics obtained through comparing between the two genotypes at each time point by Welch's *t*-test. The raw data, experimental conditions, and detailed statistics are available in *Figure 7—source data 7*. (**H**) Box plots showing ciliary signal intensity of ADCY3 in isolated hippocampal neurons prepared from *Ncs1*$^{+/+}$ or *Ncs1*$^{-/-}$ E18.5 mouse embryos at DIV7. The cells were fixed and stained with α-ADCY3 antibody, and imaged via wide-field microscopy. Data are averaged from five different neurons per genotype. Statistical significance was calculated from nested *t*-test. The raw data, experimental conditions, and detailed statistics are available in *Figure 7—source data 8*. A.U., arbitrary units; n.s., not significant; *p < 0.05, **p < 0.01.

The online version of this article includes the following source data and figure supplement(s) for figure 7:

**Source data 1.** Immunofluorescence conditions in the experiment shown in *Figure 7A*.

**Source data 2.** Immunofluorescence conditions in the experiment shown in *Figure 7B*.

**Source data 3.** Immunofluorescence conditions in the experiment shown in *Figure 7C*.

**Source data 4.** Immunofluorescence conditions in the experiment shown in *Figure 7D*.

**Source data 5.** Raw quantification data, immunofluorescence conditions, and detailed statistics of the experiment shown in *Figure 7E*.

**Source data 6.** Raw quantification data, immunofluorescence conditions, and detailed statistics of the experiment shown in *Figure 7F*.

**Source data 7.** Raw quantification data, immunofluorescence conditions, and detailed statistics of the experiment shown in *Figure 7G*.

**Source data 8.** Raw quantification data, immunofluorescence conditions, and detailed statistics of the experiment shown in *Figure 7H*.

**Figure supplement 1.** (**A**) The expression of NCS1 in various tissues.

**Figure supplement 1—source data 1.** The original files of the full raw unedited blots shown in *Figure 7—figure supplement 1A*.

**Figure supplement 1—source data 2.** The uncropped blots with boxes that indicate the regions displayed in *Figure 7—figure supplement 1A*.

**Figure supplement 2.** NCS1 localizes to the ciliary base in most ciliated tissues but not in photoreceptor cells.

**Figure supplement 2—source data 1.** Immunofluorescence conditions in the experiment shown in *Figure 7—figure supplement 2A–E*.

**Figure supplement 3.** Localization of ciliary GPCRs is mildly decreased in hippocampal neurons prepared from *Ncs1*$^{-/-}$ mice.

**Figure supplement 3—source data 1.** Immunofluorescence conditions, raw quantification data, and detailed statistics of the experiment shown in *Figure 7—figure supplement 3A*.

**Figure supplement 3—source data 2.** Immunofluorescence conditions, raw quantification data, and detailed statistics of the experiment shown in *Figure 7—figure supplement 3B*.

## *Ncs1* knockout mice exhibit obesity, but not other phenotypes related to ciliopathies

To date, NCS1 has been characterized mainly in neuronal aspects, as NCS1 was classically believed to be a neuron-specific calcium sensor (*Olafsson et al., 1997*). Ncs1 was shown to be essential for memory formation in *C. elegans*. Mice lacking Ncs1 exhibited impairment of memory formation (*de Rezende et al., 2014*; *Saab et al., 2009*; *Nakamura et al., 2017*). Since NCS1 is now shown to localize to the distal appendage in both neuronal and non-neuronal cells and regulate efficient cilium formation at least in some cell types, we sought to test if *Ncs1* knockout mice show phenotypes related to ciliopathies, pleiotropic disorders caused by functional and structural dysfunction of cilia (*Reiter and Leroux, 2017*). A series of previous genetic studies in mice showed that the loss of ciliary function results in a variety of disorders ranging from developmental defects, including neural tube defect, skeletal anomalies as well as left–right patterning defects, to obesity, retinal degeneration, cystic kidney diseases, liver fibrosis, and male infertility (*Norris and Grimes, 2012*). The phenotypes found in ciliopathy mouse models greatly vary depending on the timing of gene deletion and which ciliopathy gene is mutated in the model, likely reflecting the differences in the severity of the defects in cilium formation and function, as well as the cell types that the gene mutations affect. We

assessed whether the previously generated *Ncs1*⁻/⁻ mice (*Hermainski, 2012*; *Ng et al., 2016*) exhibit ciliopathy phenotypes. Inconsistent with the previous two reports (*Nakamura et al., 2011*; *Dickinson et al., 2016*), which generated *Ncs1* knockout mice independently, our *Ncs1*⁻/⁻ mice did not exhibit preweaning lethality (*Figure 7A*, p = 0.369 in Chi-square test in data with male and female combined). The difference between our data and the previous studies might derive from the background of mice (C57BL/6J in our study and C57BL/6N in the previous studies). When body weight was analyzed, both male and female *Ncs1*⁻/⁻ mice became more obese than their littermates starting at 9–10 weeks of age and gained 10% more weight than the controls at 20 weeks (*Figure 8B, C*). The obesity phenotype is consistent with the previous reports (*Nakamura et al., 2011*; *Ratai et al., 2019*) and is similar to what was observed in cilia-defective mice, which became obese starting between 8 and 12 weeks (*Ding et al., 2020*; *Fath et al., 2005*; *Mykytyn et al., 2004*; *Nishimura et al., 2004*). *Ncs1*⁻/⁻ accumulated more fat than their littermate *Ncs1*⁺/⁻ mice (*Figure 8D*), suggesting that the obesity phenotype at least partially comes from the increased fat amount in *Ncs1*⁻/⁻ mice. We also assessed other ciliopathy phenotypes, but *Ncs1*⁻/⁻ did not show other cilia-related symptoms, such as retinal degeneration (judged by thickness of outer nuclear layer), polycystic kidney, and male infertility (*Figure 8E–H*). The absence of retinal degeneration, one of the most penetrant phenotypes besides obesity in Bardet–Biedl syndrome (*Forsythe and Beales, 2013*; *Forsyth and Gunay-Aygun, 2020*), might reflect the lack of Ncs1 at the ciliary base in photoreceptors (*Figure 7—figure supplement 2E*). The milder phenotype of *Ncs1*⁻/⁻ mice than the previously reported cilia-defective mice may reflect mild-modest cilium formation defect of *Ncs1*⁻/⁻ mice (*Figure 7E, G*). Further investigations are needed to determine whether the obesity phenotype singularly come from cilia defect, and how exactly dysfunction of cilia leads to obesity in *Ncs1*⁻/⁻ mice.

## Discussion

In 1962, Sorokin described through extensive electron microscopy that cilium biogenesis in fibroblasts is initiated by attachment of a vesicle to the distal end of the centriole (*Sorokin, 1962*), or more precisely to the distal appendage of the mother centriole (*Schmidt et al., 2012*). In the follow-up study in 1968, Sorokin observed smaller vesicles that may be attached to a single blade of the distal appendages (*Sorokin, 1968*). These small vesicles were named as the distal appendage vesicles by Westlake group (*Lu et al., 2015*). We believe these distal appendage vesicles share substantial overlap with the RAB34 vesicles defined here; however, the shape of RAB34-positive vesicles appear to be highly variable even before the cilium formation was induced by serum starvation (Figure 3—figure supplement 3G–J of the *Kanie et al., 2025*), where the membrane fusion factors, EHD1 and PACSIN2, were not recruited (Figure 3C, D of *Kanie et al., 2025*). This may suggest that RAB34-positive vesicles at the mother centriole may fuse to form vesicles with a variety of sizes or different sized vesicles are recruited to the distal appendages. Thus, the RAB34-positive vesicles described in this study may be different from what were proposed as the distal appendage vesicles. The sum of these studies emphasizes that we need to accumulate more knowledge to accurately define these vesicles.

The study from Westlake group also suggested that the distal appendage vesicles may be derived from RAB11-positive preciliary vesicles (*Lu et al., 2015*), which are critical for bringing RAB8 to the ciliary membrane (*Westlake et al., 2011*). However, RAB8 is only recruited at the later stage of the cilium biogenesis (*Lu et al., 2015*), and there is no direct evidence showing that RAB11-positive vesicles are captured by the distal appendages. Nevertheless, precursor vesicles, termed as "preciliary vesicles", are budded from membranous organelles, then trafficked to the centriole, and finally captured by distal appendage proteins to become the distal appendage vesicles.

Since the distal appendage proteins that have been discovered so far CEP83 (*Tanos et al., 2013*), CEP164 (*Graser et al., 2007*), TTBK2 (*Čajánek and Nigg, 2014*), SCLT1 (*Tanos et al., 2013*), FBF1 (*Tanos et al., 2013*), CEP89 (*Sillibourne et al., 2011*), ANKRD26 (*Bowler et al., 2019*), LRRC45 (*Kurtulmus et al., 2018*) lack apparent lipid-binding motifs, how exactly the preciliary vesicle is captured by the distal appendage is poorly understood. In an accompanying paper, we screened all the previously and newly discovered distal appendage proteins and found that CEP89 is important for preciliary vesicle recruitment but not for other processes of cilium formation, such as IFT and CEP19 recruitment (see Figure 5 of *Kanie et al., 2025*). Since CEP89 also lacks any identifiable lipid-binding domain, we hypothesized that an interactor of CEP89 would be directly involved in the preciliary

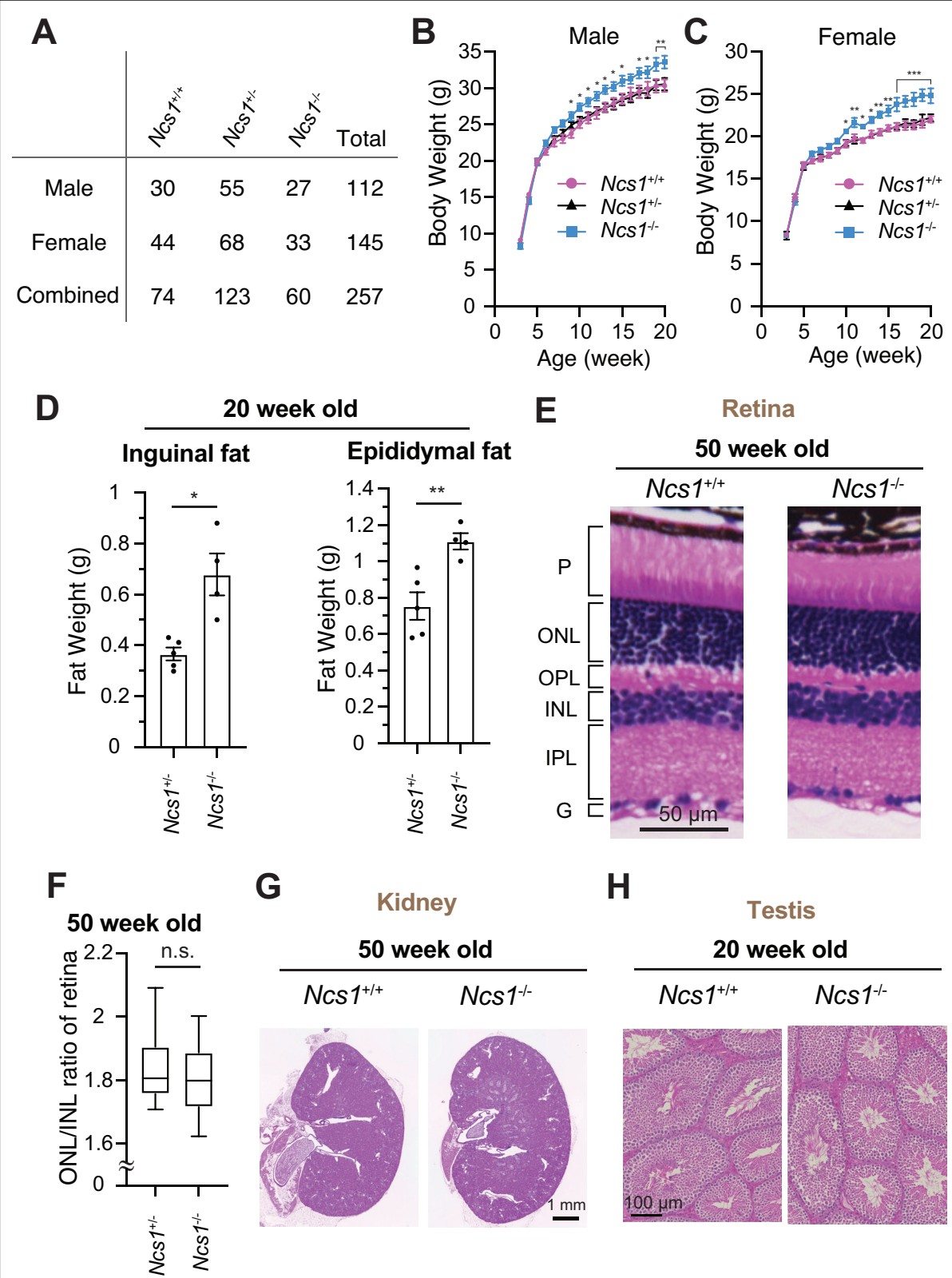

**Figure 8.** *Ncs1* knockout mice display obesity but no other ciliopathy-related phenotypes. (**A**) A table showing viability of *Ncs1*⁺/⁺, *Ncs1*⁺/⁻, or *Ncs1*⁻/⁻ mice, generated by crossing *Ncs1*⁺/⁻ male and female, at P21. Detailed information of the mice and statistics are available in *Figure 8—source data 1*. Body weight analysis of *Ncs1*⁺/⁺, *Ncs1*⁺/⁻, or *Ncs1*⁻/⁻ male (**B**) and female (**C**) mice. Raw data and detailed statistics are available from *Figure 8—source data 2*. (**D**) Measurements of the weights of inguinal fat (left) or epididymal fat (right) from 20-week-old *Ncs1*⁺/⁺ or *Ncs1*⁻/⁻ male mice. Raw data and

*Figure 8 continued on next page*

*Figure 8 continued*

detailed statistics are available in *Figure 8—source data 3*. (**E**) Hematoxylin and eosin (H&E) staining of the retina prepared from 50-week-old *Ncs1$^{+/+}$* or *Ncs1$^{-/-}$* female mice. Scale bar: 50 μm. G, ganglion cell layer; IPL, inner plexiform layer; INL, inner nuclear layer; OPL, outer plexiform layer; ONL, outer nuclear layer; P, photoreceptor cell layer. Representative images from five *Ncs1$^{+/+}$* or *Ncs1$^{-/-}$* mice are shown. Detailed information of the mice is available in *Figure 8—source data 4*. (**F**) Quantification of ONL/INL ration of the retina prepared from 50-week-old *Ncs1$^{+/+}$* or *Ncs1$^{-/-}$* mice. 8 areas per mouse and 5 mice from each genotype were analyzed. Statistical significance was calculated from nested *t*-test. The raw data, detailed information of the mice, and detailed statistics are available in *Figure 8—source data 5*. H&E staining of the kidney (**G**) or testis (**H**) prepared from 50-week-old *Ncs1$^{+/+}$* or *Ncs1$^{-/-}$* female mice (**G**) or 20-week-old *Ncs1$^{+/+}$* or *Ncs1$^{-/-}$* male mice (**H**). Scale bar: 1 mm (**G**) and 100 μm (**H**). Representative images from five (**G**) or three (**H**) *Ncs1$^{+/+}$* or *Ncs1$^{-/-}$* mice are shown. Detailed information of the mice is available in *Figure 8—source data 4*. n.s., not significant; *p < 0.05, **p < 0.01, ***p < 0.001.

The online version of this article includes the following source data for figure 8:

**Source data 1.** Detailed information of the mice and detailed statistics of the experiment shown in *Figure 8A*.

**Source data 2.** Raw data and detailed statistics of the experiment shown in *Figure 8B, C*.

**Source data 3.** Raw data and detailed statistics of the experiment shown in *Figure 8D*.

**Source data 4.** Information of the mice used in the experiments shown in *Figure 8E, G, H*.

**Source data 5.** Raw data and detailed statistics of the experiment shown in *Figure 8F*.

vesicle recruitment. In this paper, we discovered NCS1 as a stoichiometric interactor of CEP89. We further show that NCS1 captures the preciliary vesicle via its myristoylation motif.

## How NCS1 captures preciliary vesicles only at distal appendages?

To make cilium formation efficient and error-free, we assume that the cells would have mechanisms where NCS1 only captures the preciliary vesicle at the distal appendage but not at other locations within cells (e.g., cytoplasm). In addition to its centriolar localization, NCS1 localizes throughout cytoplasm, of which signal is completely lost in *NCS1* knockout cells (*Figure 2A*). This implies that NCS1 may sequester its myristoylation motif to remain in the cytoplasm and may expose the membrane association motif to capture the preciliary vesicle only at the distal appendage. One very intriguing possibility is that NCS1 may extend its myristoyl group in response to increases in local calcium concentration (calcium-myristoyl switch), as shown for other NCS family proteins, such as Recoverin and Hippocalcin (*Ames et al., 1997*; *O'Callaghan et al., 2003*). The local calcium concentration may be higher at the centriole because of the high calcium concentration in the cilium (*Delling et al., 2013*). We addressed this question by making a series of EF-hand mutants of NCS1, where the amino acids at the -z position required for calcium binding were mutated. Our data emphasize the importance of calcium in the stability of NCS1 (*Figure 5—figure supplement 1A*). As long as the expression level of NCS1 is maintained, the mutation did not strongly affect either centriolar localization or cilium formation (*Figure 5—figure supplement 1B, C*). These data suggest that calcium may be required for the structural integrity of NCS1 but may not regulate protrusion of myristoyl moiety, which is required for efficient preciliary vesicle recruitment and subsequent cilium formation (*Figure 5E, F*). Our data align with the previous reports that suggest the absence of calcium-myristoyl switch in NCS1 (*Ames et al., 2000*; *O'Callaghan et al., 2002*; *Lemire et al., 2016*). The second possibility is that NCS1 exposes its myristoyl group only when the protein binds to another protein at the distal appendage. Given that CEP89 recruits NCS1 to the distal appendage (*Figure 2L*), we wondered if NCS1 associates with membranes only when the protein binds to CEP89. However, a fractionation experiment showed that NCS1 purifies with the membrane fraction even in the absence of CEP89 (*Figure 5A*), indicating the absence of CEP89-myristoyl switch. The third possibility is that NCS1 continuously expose its myristoylation motif but remains in cytoplasm because of a weak membrane association. This hypothesis is in agreement with the low affinity of myristoylated peptides for lipid with the dissociation constant of 100 μM, which is barely sufficient to keep its membrane association (*Peitzsch and McLaughlin, 1993*). Myristoylated proteins typically require additional mechanisms to bind membranes (*Wright et al., 2010*): (1) another acyl chain (e.g., palmitoylation), (2) a cluster of basic amino acids that help association with negatively charged head group of the membrane, and (3) an interacting partner that has affinity for membrane. Since NCS1 does not appear to have another acyl chain, membrane binding of NCS1 is likely enhanced by either basic amino acids or another distal appendage protein that is in close proximity to NCS1 keeps the protein associated with membrane. Interestingly, a recent

paper showed that lysine residues at positions 3, 7, and 9 may be also involved in membrane binding of NCS1 (*Baksheeva et al., 2020*). In terms of the binding partner induced membrane association, this process is likely regulated by yet unknown distal appendage protein(s) and not by CEP89 as membrane association of NCS1 does not require CEP89 (*Figure 5A*). This is also supported by the NCS1-CEP89 structural model, which places CEP89 opposite to the myristoylated glycine and the three calcium-binding sites of NCS1 (*Figure 5H*). The rise in the local concentration of NCS1 as well as membrane vesicle at the centriole may also help NCS1's membrane association at that location. The weak association between myristoylated NCS1 and membrane could also explain why cells can capture preciliary vesicles, albeit less efficiently, even in the absence of NCS1 (*Figures 3C and 5F*). We currently do not have an obvious candidate for a distal appendage protein that may compensate the lack of NCS1, as the depletion of NCS1 in any of the knockouts of the known distal appendage proteins did not further inhibit the preciliary vesicle recruitment and cilium formation (*Figure 4B–D*). Future studies will focus on identifying additional protein(s) that recruit the preciliary vesicle to the centriole.

## How the preciliary vesicle is transported to the distal appendage

Our model suggests that the preciliary vesicle is recruited to the distal appendage in a microtubule-dependent manner, whereas NCS1 can reach to the mother centriole without intact microtubules (*Figure 6*). How is the preciliary vesicle recruited to the distal appendage? Classically, subdistal appendages were considered as the site where the microtubule anchoring occurs, as the electron micrograph showed that microtubule are in contact with the head of the subdistal appendage (see Figure 12 of the *Vorobjev and Chentsov Yu, 1982*). The subdistal appendage localization of Ninein, which was shown to be indispensable for microtubule anchoring at the centriole (*Delgehyr et al., 2005*), further supports that subdistal appendages are the contact site of the microtubule. Therefore, one can hypothesize that the preciliary vesicle is transported first to the subdistal appendage and then subsequently moves to the distal appendage by an unknown mechanism. Interestingly, CEP89 and its interactors, NCS1 and CEP15, each localize to positions near the subdistal appendage in addition to their distal appendage localization (*Figure 2D, F*; *Chong et al., 2020*). NCS1 may bind to the preciliary vesicle at the subdistal appendage and then move to the distal appendage to anchor the vesicle and promote cilium formation. Possibly this process is rapid, so that any vesicle attached to the subdistal appendage has never been observed in electron micrographs. However, this model conflicts with the observation that subdistal appendages are dispensable for cilium formation (*Mazo et al., 2016*). Alternatively, microtubules may populate a structural site around the distal appendages as shown by recent dSTORM imaging (*Chong et al., 2020*). γ-Tubulin observed in vicinity of the distal appendage may nucleate those microtubules. If this is the case, the preciliary vesicle may be transported directly to the distal appendage and then be captured by NCS1. To address this question, it would be greatly informative if the entire preciliary vesicle recruitment process could be visualized by super-resolution microscopy in live cells in a future study. Another important question is whether NCS1 specifically recognizes a receptor on the preciliary vesicle or NCS1 randomly captures the membrane of vesicles that arrive at the distal appendage. NCS1 may recognize specific vesicles via the membrane curvature or specific lipid components. Interestingly, a recent study showed that NCS1 preferentially binds to phosphatidylinositol-3-phosphate (*Baksheeva et al., 2020*). This warrants future study.

## Requirement of NCS1 in cilium formation differs among cell types

Cilium formation can be classified into two types (*Sorokin, 1968*; *Molla-Herman et al., 2010*): (1) the intracellular pathway, which is initiated by preciliary vesicle recruitment to the distal appendage, and (2) the extracellular pathway, where the centriole first docks to plasma membrane. While specific cell types have been observed to selectively use one of the two pathways, the distinction between the pathways might not be so definitive. For example, mouse inner medullary collecting duct cells (mIMCD3), typically classified as using the extracellular pathway, can use the intracellular pathway in less confluent cells (*Stuck et al., 2021*). Nonetheless, the requirement for RAB34,the centriole-associated vesicle marker, in ciliogenesis is more pronounced in the cells that use the intracellular pathway (*Ganga et al., 2021*; *Oguchi et al., 2020*; *Stuck et al., 2021*), indicating that preciliary vesicle recruitment is an indispensable step for that pathway. Our data showed that cilium formation is modestly affected by NCS1 depletion in the cell types that are known to use intracellular pathway

(*Molla-Herman et al., 2010*), such as RPE and MEFs (*Figures 3A and 7E*). In contrast, we did not see apparent cilia formation defects in primary neurons isolated from E18.5 mice (*Figure 7G*). A possible explanation for this is that the neurons use the extracellular pathway, however, the ciliogenesis pathway for neurons is not well characterized. The papers reported the presence of the ciliary pocket, a sign of the intracellular pathway (*Molla-Herman et al., 2010*), in electron micrographs of Grueneberg ganglion neurons from young mice (P15) (*Brechbühl et al., 2008*) and neural progenitors (*Breunig et al., 2008*; *Han et al., 2008*; *Mirzadeh et al., 2008*). Recent volume electron microscopy studies, however, suggested that cortical and hippocampal neurons from adult animals typically do not possess apparent ciliary pocket (*Sheu et al., 2022*; *Wu et al., 2024*). Thus, the ciliogenesis pathways in neurons may differ depending on subtypes or during development. While we do not know the ciliogenesis pathway that our primary hippocampal neurons used, the lack of cilium formation defect may be because the cells use the extracellular pathway. Future studies will focus on determining the requirement of NCS1 in the extracellular pathway. Another possible explanation for a failure to see cilium formation defects is that it is not easy to assess the kinetics of ciliation in isolated hippocampal neurons because culture conditions are very different from RPE cells. Notably, cilium formation is not induced by serum starvation in isolated hippocampal neurons. In tissues in vivo, it was not easy to assess whether the $Ncs1^{-/-}$ mice have fewer cilia than the control mice for several reasons. First, cilium structure is greatly affected by sample preparation. For example, we cannot visualize cilia if we do not fix the tissues by cardiac perfusion with 4% paraformaldehyde (PFA) and it is difficult to achieve perfectly efficient perfusion. Second, orientation of cilia is affected by the orientation of how the tissue is sectioned and it is thus difficult to analyze cilia that elongate perpendicularly to the slice. Therefore, we could not test whether NCS1 is required for cilium formation in cells that typically use the extracellular pathway. These questions warrant future studies. Importantly, we did observe a decrease in ciliary localization of several membrane proteins, such as ARL13B (*Figure 3—figure supplement 1B–D*) and ADCY3 (*Figure 7H*) in $Ncs1^{-/-}$ cells, even when the percentage of ciliated cells was comparable to the control cells. This may suggest that NCS1 might be involved in recruiting membrane signaling proteins to the cilium besides its function in preciliary vesicle recruitment and cilium formation. It would be interesting to test in the future studies if other ciliary membrane proteins are also brought to the cilium via ciliary vesicles.

### *NCS1* may be a ciliopathy gene

Given that NCS1 is involved in preciliary vesicle recruitment and subsequent cilium formation, we tested if $Ncs1^{-/-}$ mice exhibit ciliopathy phenotypes. Our data showed that $Ncs1^{-/-}$ mice display a modest obesity phenotype, but no other apparent ciliopathy-related phenotypes, including retinal degeneration. The absence of retinal degeneration may be explained by the lack of Ncs1 at the ciliary base in photoreceptors. A possible explanation for the lack of other ciliopathy phenotypes is the partial penetrance of these other symptoms. In human, obesity and retinal degeneration is observed in most Bardet–Biedl syndrome patients (~90%), whereas other phenotypes, such as hypogonadism and kidney disease are often absent (*Forsyth and Gunay-Aygun, 2020*). Mice lacking the distal appendage protein, FBF1 (*Zhang et al., 2021*) or ANKRD26 (*Acs et al., 2015*; *Bera et al., 2008*), or the distal appendage associate protein CEP19 (*Shalata et al., 2013*), display morbid obesity with few other ciliopathy-related phenotypes (e.g., preweaning lethality and hydrocephalus in $Fbf1^{-/-}$ and male infertility in $Cep19^{-/-}$ mice). Interestingly, our data reveal that knockouts of each of these genes in RPE1-hTERT cells show a kinetic defect in ciliation, but the cells eventually catch up to complete cilium formation (Figure 5A, B in *Kanie et al., 2025* for ANKRD26 and FBF1 Figure 3C of *Kanie et al., 2017* for CEP19). This phenotype is almost identical to that observed in *CEP89* or *NCS1* knockout cells (*Figure 3A*). This suggests that quantitative defects in cilium formation defect may drive obesity with few other ciliopathy-related defects. Another explanation for the lack of other ciliopathy phenotypes besides obesity is the background of our $Ncs1^{-/-}$ mice (C57BL/6J). It is well known that depletion of the same gene could cause different severity of the phenotypes in different background of mice. For example, mice lacking Bbip1, a BBSome-associated protein (*Loktev et al., 2008*), in pure C57BL/6J background show complete perinatal lethality, while approximately half of the $Bbip1^{-/-}$ mice in 129/SvJ background can survive into adulthood (*Loktev and Jackson, 2013*). Interestingly, two independent reports showed that $Ncs1^{-/-}$ exhibit partial (~50%) preweaning lethality (*Dickinson et al., 2016*; *Nakamura et al., 2011*) in C57BL/6N mice. The difference in the severity of the phenotypes in

*Ncs1⁻/⁻* mice between previous reports and our results may be explained by the difference between C57BL/6J and C57BL/6N. Genetic and phenotypic differences between these two strains were extensively described in the previous paper (*Simon et al., 2013*). Thus, NCS1 may be a ciliopathy gene and obesity caused by NCS1 depletion may be attributable to ciliary defect. This warrants future genetic study. If obesity accompanied with NCS1 depletion is due to a cilia defect, what kind of cilia defect exist in the *Ncs1* defective animals in vivo? A simple defect may be the reduced number of ciliated cells because of the cilium formation defect. While we did not see an apparent decrease in the number of cilia in *Ncs1⁻/⁻* mice in any tissues that we examined (e.g., brain, kidney, pancreatic islets, and airway epithelia) (*Figure 7B–D*, *Figure 7—figure supplement 2*), more accurate characterization is needed to make a conclusion. It is possible that cilium formation is abolished in developmentally and spatially regulated manner, so that the defect may be only apparent in specific cell types and developmental stage. Another possibility that may cause cilia-related obesity phenotype in *Ncs1⁻/⁻* mice is that localization of some of the ciliary membrane proteins may be abolished in *Ncs1⁻/⁻* cells as shown in the cultured hippocampal neurons (*Figure 7H*). Unfortunately, it is not easy to assess the number and morphology of the cilia as well as signal intensity of the ciliary membrane proteins in vivo because of the issues described above. Technical improvement in the future may allow us to more accurately characterize the cilia in vivo and determine whether ciliary defects in *Ncs1⁻/⁻* mice indeed cause obesity. Alternatively, it would be interesting to see if *Cep89* knockout mice display the similar phenotypes as *Ncs1* knockout mice.

## The connection between NCS1-related neurological disorder and cilia defect

NCS1 has been shown to participate in memory formation in *C. elegans* (*Gomez et al., 2001*) and mice (*Saab et al., 2009*; *Nguyen et al., 2021*; *Nakamura et al., 2017*; *Ng et al., 2016*; *de Rezende et al., 2014*). While the neurological phenotypes in *Ncs1⁻/⁻* mice are not consistent across studies, possibly because of the differences in mouse background, it seems that many studies agree that the overall phenotypes are mild, and the mice display defects in memory formation, when tested for novel object recognition (*de Rezende et al., 2014*) or displaced object recognition (*Mun et al., 2015*; *Ng et al., 2020*; *Nguyen et al., 2021*). It is intriguing to consider whether the memory formation defect in *Ncs1⁻/⁻* mice is attributable to ciliary defects. Several lines of evidence suggest that loss of cilia in brain results in memory formation defects. If IFT88, an IFT component critical for formation of the cilium, is depleted in telencephalon by Emx1-Cre, the mice display impaired recognition memory assessed through novel object recognition test (*Berbari et al., 2014*). The depletion of IFT20 in dentate gyrus of the hippocampus using AAV-CAMKII-Cre caused the defect in displaced object recognition test (*Rhee et al., 2016*). Both mice lacking ADCY3 or SSTR3, ciliary membrane proteins that are prominent in neurons (*Bishop et al., 2007*), exhibit defect in novel object recognition (*Einstein et al., 2010*; *Wang et al., 2011*). The similarity between cilia-defective mice and *Ncs1⁻/⁻* mice may suggest that the memory formation defect in *Ncs1⁻/⁻*-deficient mice may be due to ciliary dysfunction. It would be interesting to test whether SSTR3 agonist, which induces long-term potentiation (LTP) (*Einstein et al., 2010*) likely via binding to the ciliary G-protein-coupled receptor, SSTR3, can induce LTP in *Ncs1⁻/⁻* mice. It would be also interesting to see if Cep89 depletion in mice causes similar memory formation defect, since *NCS1* knockouts and *CEP89* knockouts showed almost identical cilium formation defects (*Figure 3*). The importance of cilia in neurological deficiencies should be an area of extensive future study.

## Materials and methods

### Plasmids

pMCB306, a lenti-viral vector containing loxP-mU6-sgRNAs-puro resistance-EGFP-loxP cassette, and P293 Cas9-Blue Fluorescent Protein (BFP) were gifts from Prof. Michael Bassik. Lenti-virus envelope and packaging vector, pCMV-VSV-G and pCMV-dR8.2 dvpr, respectively, were gifts from Prof. Bob Weinberg (Addgene plasmid #8454 and #8455).

pOG44 (V600520) was obtained from Thermo Fisher Scientific.

Lenti-viral vectors containing single-guide RNAs (sgRNAs) were generated by ligating 50 fmol oligonucleotides encoding sgRNAs into 25 ng of the pMCB306 vector digested with BstXI (R0113S,

NEB) and BlpI (R0585S, NEB) restriction enzymes along with 0.25 µl of T4 ligase (M0202S, NEB) in 2.5 µl total reaction volume. Before ligation, 4 µM of forward and reverse oligonucleotides listed in *Source data 1* were annealed in 50 µl of annealing buffer (100 mM potassium acetate, 30 mM HEPES (pH7.4), and 3 mM magnesium acetate) at room temperature following denaturation in the same buffer at 95°C for 5 min. The targeting sequence for sgRNAs is listed in *Source data 1*. The guide RNA targeting sequence for pMCB306-sgNCS1 vector used to create cells lacking both NCS1 and each of the other distal appendage proteins shown in *Figure 4* is the same as the one used to make *NCS1* knockout cells. The knockout cells for other distal appendage proteins were described in an accompanying paper (*Kanie et al., 2025*).

pG-LAP6/puro vector (pCDNA5/TO/FRT/EGFP-TEV cleavage site-S tag-PreScission cleavage site/DEST) used for the tandem affinity purification experiment was previously described (*Kanie et al., 2017*). Gateway cloning compatible lenti-viral vectors, pWPXLd/LAP-N/puro/DEST vector and pWPXLd/LAP-C/puro/DEST vector, were previously described (*Kanie et al., 2017*). pWPXLd/LAP-N/blast/long EF/DEST was created by inserting N-terminally LAP tag (EGFP-TEV cleavage site-S tag-PreScission cleavage site)/DEST/blasticidin resistance cassette into a second generation lenti-viral vector, pWPXLd. pWPXLd vector was a gift from Prof. Didier Trono (Addgene plasmid #12258). pWPXLd/LAP-C/blast/long EF/DEST vector was created by inserting DEST/C-terminally LAP tag/blasticidin resistance cassette into the pWPXLd vector. pWPXLd/FLAG-N/blast/DEST vector was created by inserting FLAG/DEST/blasticidin resistance cassette into the pWPXLd vector. All the lenti-viral vectors were propagated in Stbl3 competent cells to reduce unwanted recombination of long terminal repeat of the vectors.

pCS2-N-terminal 5×MYC/DEST and pCS2-N-terminal 3×HA/DEST (used for in vitro translation) were created by inserting either 5×MYC tag or 3×HA tag and destination cassette into pCS2+ vector, which contains Sp6 and CMV promoter.

The Gateway entry vector for *Homo sapiens* CEP89 was created by BP recombination using a polymerase chain reaction (PCR) product containing attB1 and attB2 sites, which was amplified using pCR4-TOPO-CEP89 (MHS6278-213243472, Open Biosystems) as a template. Gateway entry vectors carrying truncation mutants of CEP89 (1–343 a.a. and 344–783 a.a.) were created by using BP recombination between pDONR221 and PCR-amplified inserts.

The Gateway entry vectors for *H. sapiens* NCS1 (HsCD00366520) and CEP15 (HsCD00365881) were obtained from Harvard plasmid. STOP codons were added or removed by using Quick change mutagenesis if necessary. The Gateway entry vectors for NCS1 mutants (myristoylation defective or EF-hand mutants) were created via Quick change mutagenesis using the entry vector for NCS1 described above. The quick change mutagenesis was performed by PCR with a complementary primer set (forward and reverse) that has a point mutation in the middle of the primers. Following the PCR, the PCR product was treated with 20U of DpnI (R0176L, NEB) for 1 hr at 37°C to eliminate the template, and was then used to transform competent cells.

The entry vectors for the CEP350 fragment (2470–2836 a.a.) and FGFR1OP (or FOP) was previously described (*Kanie et al., 2017*).

Flp-In system compatible N-terminally LAP-tagged CEP89 was generated by LR recombination between CEP89 entry vector and pG-LAP6/puro.

Lenti-viral vector containing untagged CEP89 (minimal CMV promoter) was created by LR recombination between CEP89 entry vector that contains a stop codon and pWPXLd/LAPC/blast/minimal CMV/DEST vector.

Lenti-viral vectors containing untagged NCS1 (long or short EF promoter) were created by LR recombination between NCS1 (wild-type and mutants) entry vectors that contain stop codons and pWPXLd/LAPC/blast/long EF/DEST or pWPXLd/LAPC/blast/short EF/DEST vectors.

N-terminally HA-tagged CEP89, CEP15, and FGFR1OP (or FOP) vectors used for in vitro binding assay were created by LR recombination between the respective entry vectors containing a stop codon and the pCS2-N-terminal 3×HA/DEST vector. pCS2-N-terminal 5×MYC-tagged CEP15, NCS1, and the CEP350 fragment (2470–2836 a.a.) vectors were created by LR recombination between the respective entry vectors that contain stop codons and the pCS2-N-terminal 5×MYC/DEST vector.

## Cell line, cell culture, transfection, and lenti-viral expression

hTERT RPE-1 cells and 293T cells were grown in DMEM/F-12 (12400024, Thermo Fisher Scientific) supplemented with 10% FBS (100-106, Gemini), 1× GlutaMax (35050-079, Thermo Fisher Scientific), 100 U/ml penicillin–streptomycin (15140163, Thermo Fisher Scientific) at 37°C in 5% $CO_2$. To induce cilium formation, cells were incubated in DMEM/F-12 supplemented with 1× GlutaMax and 100 U/ml penicillin–streptomycin (serum-free media). Both cell lines were authenticated via a short-tandem-repeat based test. The authentication was performed by MTCRO-COBRE Cell line authentication core of the University of Oklahoma Health Science Center. Mycoplasma negativity of the original cell lines (hTERT RPE-1 and 293T) grown in antibiotics-free media was confirmed by a PCR-based test (G238, Applied Biological Materials).

RPE-FRT9 expressing N-terminally LAP-tagged CEP89 used for tandem affinity purification was generated by transfecting 150 ng of the preceding vectors with 850 ng of pOG44, followed by selection with 10 µg/ml puromycin. Flp-In system compatible RPE cells (RPE-FRT9) were previously described (*Sang et al., 2011*).

All other stable cell lines, including CRISPR knockout cells, were generated using lenti-virus. Lenti-virus carrying either gene of interest or sgRNAs was produced by co-transfecting 293T cells with 150 ng of pCMV-VSV-G, 350 ng of pCMV-dR8.2 dvpr, and 500 ng of lenti-viral transfer plasmids previously described along with 3 µl of Fugene 6 (E2692, Promega) transfection reagent. Media was replaced 24 hr after transfection to omit transfection reagent, and virus was harvested at 48 hr post-transfection. Virus was then filtered with a 0.45-µm PVDF filter (SLHV013SL, Millipore) and mixed with fourfold volume of fresh media containing 12.5 µg/ml polybrene (TR-1003-G, Millipore). Following infection for 66 hr, cells were selected with either 10 µg/ml puromycin (P9620, Sigma-Aldrich) or 10 µg/ml blasticidin (30-100-RB, Corning) for at least 10 days before subsequent analysis.

## CRISPR knockout

RPE cells expressing BFP-Cas9 were generated by infection with lenti-virus carrying P293 Cas9-BFP, followed by sorting BFP-positive cells using FACSAria (BD). RPE-BFP-Cas9 cells were then infected with lenti-virus carrying sgRNAs in the pMCB306 vector to generate knockout cells. After selection with 10 µg/ml puromycin, cells were subjected to immunoblotting, IF, or genomic PCR combined with TIDE analysis (*Brinkman et al., 2014*) to determine knockout efficiency. The exact assay used for each cell line is listed in *Source data 7*. Cells were then infected with adenovirus carrying Cre-recombinase (1045N, Vector Biolabs) at a multiplicity of infection of 50 to remove the sgRNA-puromycin resistance-EGFP cassette. Ten days after adenovirus infection, GFP-negative single cells were sorted using FACSAria. The single-cell clones were expanded, and their knockout efficiency were determined by IF, western blot, and/or genomic. The same number of validated single clones (typically three to four different clones) was mixed to create pooled single-cell knockout clones to minimize the phenotypic variability occurred in single-cell clones. The cells lacking both NCS1 and each of the other distal appendage proteins shown in *Figure 4* were created by infecting the knockout cells with lenti-virus carrying sgNCS1. The experiments shown in *Figure 4* were performed without removing loxP-mU6-sgRNAs-puro resistance-EGFP-loxP cassette.

Cells used in the rescue experiments shown in *Figure 3B*, *Figure 3—figure supplement 1A–C*, *Figure 5A–F*, and *Figure 5—figure supplement 1A–C* were created by infecting the respective knockout cells with lenti-virus carrying untagged CEP89 or NCS1 (wild-type or mutants). To rescue the ciliation defect of *CEP89* knockout cells, the expression level of CEP89 was carefully adjusted by using minimal CMV promoter to mimic endogenous CEP89 expression, since overexpression of CEP89 under the control of long EF promoter significantly inhibited cilium formation (data not shown).

## Tandem affinity purification

5 ml packed cell volume of RPE-FRT9 cells expressing N-terminally LAP-tagged CEP89 were re-suspended with 20 ml of LAP-resuspension buffer (300 mM KCl, 50 mM HEPES-KOH [pH 7.4], 1 mM EGTA, 1 mM $MgCl_2$, 10% glycerol, 0.5 mM dithiothreitol (DTT), and protease inhibitors [PI88266, Thermo Scientific]), lysed by gradually adding 600 µl 10% NP-40 to a final concentration of 0.3%, then incubated on ice for 10 min. The lysate was first centrifuged at 14,000 rpm (27,000 × $g$) at 4°C for 10 min, and the resulting supernatant was centrifuged at 43,000 rpm (100,000 × $g$) for 1 hr at 4°C to further clarify the lysate. High speed supernatant was mixed with 500 µl of GFP-coupled beads (*Torres*

*et al., 2009*) and rotated for 1 hr at 4°C to capture GFP-tagged proteins, and washed five times with 1 ml LAP200N buffer (200 mM KCl, 50 mM HEPES-KOH [pH 7.4], 1 mM EGTA, 1 mM MgCl₂, 10% glycerol, 0.5 mM DTT, protease inhibitors, and 0.05% NP-40). After re-suspending the beads with 1 ml LAP200N buffer lacking DTT and protease inhibitors, the GFP-tag was cleaved by adding 5 µg of TEV protease and rotating tubes at 4°C overnight. All subsequent steps until the cutting of bands from protein gels were performed in a laminar flow hood. TEV-eluted supernatant was added to 100 µl of S-protein agarose (69704-3, EMD Millipore) to capture S-tagged protein. After washing three times with LAP200N buffer lacking DTT and twice with LAP100 buffer (100 mM KCl, 50 mM HEPES-KOH [pH 7.4], 1 mM EGTA, 1 mM MgCl₂, and 10% glycerol), purified protein complexes were eluted with 50 µl of 2× lithium dodecyl sulfate (LDS) buffer (212 mM Tris–HCl, 282 mM Tris-base, 4% LDS, 20% glycerol, 1.02 mM EDTA, 0.13% Brilliant Blue G250, 0.05% phenol red buffer) containing 10% DTT and boiled at 95°C for 3 min. Samples were then run on Bolt Bis-Tris Plus Gels (NW04120BOX, Thermo Fisher Scientific) in Bolt MES SDS Running Buffer (B000202, Thermo Fisher Scientific). Gels were fixed in 100 ml of fixing solution (50% methanol, 10% acetic acid in Optima LC/MS grade water [W6-1, Thermo Fisher Scientific]) at room temperature, and stained with Colloidal Blue Staining Kit (LC6025, Thermo Fisher Scientific). After the buffer was replaced with Optima water, the bands were cut into eight pieces, followed by washing twice with 500 µl of 50% acetonitrile in Optima water. The gel slices were then reduced and alkylated followed by destaining and in-gel digestion using 125 ng Trypsin/LysC (V5072, Promega) as previously described (*Shevchenko et al., 2006*) with the addition of Protease Max (V2071, Promega) to increase digestion efficiency. Tryptic peptides were extracted from the gel bands and dried in a speed vac. Prior to LC–MS, each sample was reconstituted in 0.1% formic acid, 2% acetonitrile, and water. NanoAcquity (Waters) LC instrument was set at a flow rate of either 300 or 450 nl/min where mobile phase A was 0.2% formic acid in water and mobile phase B was 0.2% formic acid in acetonitrile. The analytical column was in-house pulled and packed using C18 Reprosil Pur 2.4 µM (Dr. Maisch) where the I.D. was 100 µM and the column length was 20–25 cm. Peptide pools were directly injected onto the analytical column in which linear gradients (4–40% B) were of either 80 or 120 min eluting peptides into the mass spectrometer. Either the Orbitrap Elite or Orbitrap Fusion mass spectrometers were used, where a top 15 or 'fastest' MS/MS data acquisition was used, respectively. MS/MS was acquired using CID with a collisional energy of 32–35. In a typical analysis, RAW files were processed using Byonic (Protein Metrics) using 12 ppm mass accuracy limits for precursors and 0.4 Da mass accuracy limits for MS/MS spectra. MS/MS data were compared to an NCBI GenBank FASTA database containing all human proteomic isoforms with the exception of the tandem affinity bait construct sequence and common contaminant proteins. Spectral counts were assumed to have undergone fully specific proteolysis and allowing up to two missed cleavages per peptide. All data were filtered and presented at a 1% false discovery rate (*Elias and Gygi, 2007*).

## Silver staining

5 µl of samples containing LDS buffer and DTT prepared for TAP-MS described above were mixed with 0.5 µl of 500 mM iodoacetamide (0210035105, MP Biomedicals). Proteins were separated in a 4–12% Bis-Tris gel (NP0321BOX, Invitrogen), followed by fixation of the gel overnight in 50% methanol at room temperature.

The gel was impregnate with solution C (0.8% (wt/vol) silver nitrate (S6506, Sigma), 207.2 mM ammonium hydroxide (A6899, Sigma), and 18.9 mM sodium hydroxide) for 15 min, followed by rinsing with water twice. The image was then developed in solution D (0.05% citric and 0.0185% formaldehyde in Milli-Q) until intensity of the bands increase to optimal level. The reaction was then terminated by adding stop solution (45% methanol and 10% acetic acid).

## Immunoblot

For immunoblotting, cells were lysed in NP-40 lysis buffer (50 mM Tris–HCl [pH 7.5], 150 mM NaCl, 0.3% NP-40 [11332473001, Roche Applied Science]) containing 10 µg/ml LPC (leupeptin, Pepstatin A, and chymostatin) and 1% phosphatase inhibitor cocktail 2 (P5726, Sigma). Following clarification of the lysate by centrifugation at 15,000 rpm (21,000 × *g*) for 10 min, samples were mixed with 1× LDS buffer (106 mM Tris–HCl, 141 mM Tris-base, 2% LDS, 10% glycerol, 0.51 mM EDTA, 0.065% Brilliant Blue G250, 0.025% phenol red) containing 2.5% 2-mercaptoethanol (M3148, Sigma) and incubated at 95°C for 5 min. Proteins were separated in an NuPAGE Novex 4–12% Bis-Tris protein gel (WG1402BOX,

Thermo Fisher Scientific) in NuPAGE MOPS SDS running buffer (50 mM MOPS, 50 mM Tris-base, 0.1% SDS, 1 mM EDTA, pH 7.7), and transferred onto an Immobilon-FL PVDF Transfer Membrane (IPFL00010, EMD Millipore) in Towbin Buffer (25 mM Tris, 192 mM glycine, pH 8.3). Membranes were incubated in LI-COR Odyssey Blocking Buffer (NC9232238, LI-COR) for 30 min at room temperature, and then probed overnight at 4°C with the appropriate primary antibody diluted in the blocking buffer. Next, the membrane was washed 3 × 5 min in TBST buffer (20 mM Tris, 150 mM NaCl, 0.1% Tween 20, pH 7.5) at room temperature, and incubated with the appropriate IRDye antibodies (LI-COR) diluted in the blocking buffer for 30 min at room temperature. After washing three times in TBST buffer, the membrane was scanned on an Odyssey CLx Imaging System (LI-COR) and proteins were detected at wavelengths 680 and 800 nm. Primary antibodies used for immunoblotting are listed in *Source data 4*. Secondary antibodies used for immunoblotting were IRDye 800CW donkey anti-rabbit (926-32213, LI-COR) and IRDye 680CW donkey anti-mouse (926-68072, LI-COR).

## Co-immunoprecipitation

Cells were plated in a 10-cm dish and grown to confluent. Cells were then lysed with NP-40 lysis buffer (50 mM Tris–HCl [pH 7.5], 150 mM NaCl, and 0.3% NP-40) containing 10 µg/ml LPC (leupeptin, Pepstatin A, and chymostatin) and 1% phosphatase inhibitor cocktail 2 (P5726, Sigma), followed by clarification of the lysate by centrifugation at 15,000 rpm (21,000 × g) for 10 min. The protein concentration was measured by Bradford assay as described previously (see procedure B step 8 in *Kanie and Jackson, 2018*). For GFP co-immunoprecipitation shown in *Figure 1D* and *Figure 5—figure supplement 1*, the soluble fraction was incubated with Protein A beads cross-linked with rabbit anti-GFP antibody (*Torres et al., 2009*) (10 µl bed volume per 3 mg of lysate) with end-over-end rotation for 1.5 hr at 4°C. For co-immunoprecipitation with endogenous NCS1 shown in *Figure 1—figure supplement 1A*, the lysate was incubated with mouse monoclonal anti-NCS1 antibody (sc-376206, Santa Cruz) (1 µg of antibody per 4 mg of lysate) for 1 hr with end-over-end rotation. The samples were then mixed with protein A beads (20 µl bed volume) and incubated with end-over-end rotation for 1.5 hr at 4°C. After the incubation with the beads (for both GFP co-IP and NCS1 co-IP), the samples were washed five times with IP wash buffer (50 mM Tris–HCl [pH 7.5], 150 mM NaCl, and 0.1% NP-40). Samples were then eluted with 2× LDS buffer containing 2.5% 2-mercaptoethanol (M3148, Sigma).

## Subcellular fractionation

Cells were plated in a 15-cm dish at the density of $1.25 \times 10^6$ cells and grown in DMEM/F-12 media containing 10% FBS for 90 hr. Cells were detached from the plate using 0.05% trypsin/EDTA (25300-054, Gibco) and pelleted down by centrifugation at 500 × g at 4°C for 5 min. After washing once with 15 ml of ice cold low osmotic buffer (25 mM HEPES-NaOH (pH7.5), 0.5 mM MgCl₂), the cell pellet was re-suspended in 10 ml of the ice cold low osmotic buffer and incubated on ice for 10 min to let the cells swollen. The swollen cells were pelleted down and were re-suspended in 0.8 ml of the ice cold low osmotic buffer, followed by nitrogen cavitation at 300 psi for 30 min on ice. The cavitate was centrifuged at 1000 × g at 4°C for 10 min. The 1000 g supernatant was then centrifuged at 15,000 × g at 4°C for 10 min. The 15,000 g supernatant was then transferred to an ultracentrifugation tube (343778, Beckman) and ultracentrifuged at 100,000 × g ($R_{max}$) in a TLA100.2 fixed angle rotor (Beckman) at 4°C for 1 hr. The supernatant samples were prepared by mixing 25 µl of supernatant from each centrifugation speed with 25 µl of 2× LDS buffer containing 5% 2-mercaptoethanol. The pellet samples were prepared by re-suspending the pellet with appropriate amount of 1× LDS buffer containing 2.5% 2-mercaptoethanol.

## In vitro binding assay

Co-In vitro translated (co-IVT) proteins were generated with pCS2-N-terminal 5×MYC vectors and pCS2-N-terminal 3×HA vectors described above using TnT Coupled Reticulocyte Lysate System under the SP6 promoter (L4600, Promega) and by following the manufacturer's recommendations with few modifications. Briefly, instead of in vitro translating 1 µg of plasmid for each reaction, 0.5 µg of HA-tagged protein along with 0.5 µg of corresponding MYC-tagged protein was co-translated. Note that we only observed the interaction between CEP89 and NCS1 when the two proteins were co-translated. The interaction between all the other proteins was identical between original protocol and co-IVT. For each pull-down reaction, 50 µl of co-IVT protein was added along with 5 µl (bed volume)

of washed HA-beads (11815016001, Roche) in 300 µl binding buffer (25 mM HEPES-NaOH [pH 7.5], 500 mM NaCl, 1 mM CaCl$_2$, and 0.1% Triton X-100) and mixed for 2 hr at 4°C. The beads were washed five times with the same buffer and eluted with 1× LDS buffer containing 2.5% 2-mercaptoethanol. The eluates were then resolved by SDS–PAGE and analyzed by immunoblotting with anti-HA (901501, BioLegend) and anti-MYC (ab9106, Abcam) antibodies.

## Transmission electron microscopy

Either control (sgGFP) or *NCS1* knockout RPE cells were grown to confluent on 12 mm round coverslips (12-545-81, Fisher Scientific), followed by serum starvation for 3 hr. Cells were then fixed with 4% PFA (433689 M, Alfa Aesar) and 2% glutaraldehyde (G7526, Sigma) in sodium cacodylate buffer (100 mM sodium cacodylate and 2 mM CaCl$_2$, pH 7.4) for 1 hr at room temperature, followed by two washes with sodium cacodylate buffer. Cells were then post-fixed in cold/aqueous 1% osmium tetroxide (19100, Electron Microscopy Sciences) in Milli-Q water for 1 hr at 4°C, allowed to warm to room temperature for 2 hr rotating in a hood, and washed three times with Milli-Q water. The samples were then stained with 1% uranyl acetate in Milli-Q water at room temperature overnight. Next, the samples were dehydrated in graded ethanol (50%, 70%, 95%, and 100%), followed by infiltration in EMbed 812. Ultrathin serial sections (80 nm) were created using an UC7 (Leica, Wetzlar, Germany), and were picked up on formvar/carbon coated 100 mesh Cu grids, stained for 40 s in 3.5% uranyl acetate in 50% acetone followed by staining in Sato's Lead Citrate for 2 min. Electron micrographs were taken on JEOL JEM1400 (120 kV) equipped with an Orius 832 digital camera with 9 µm pixel (Gatan). To test the percentage of the vesicle positive centriole, multiple serial sections (typically 3–4) were analyzed per each mother centriole, as the vesicles are often not attached to all nine blades of the distal appendage (i.e., the vesicles are often not found in all the sections of the same mother centriole).

## Immunofluorescence

For wide-field microscopy, cells were grown on acid-washed 12 mm #1.5 round coverslips (72230-10, Electron Microscopy Sciences) and fixed either in 4% PFA (433689M, Alfa Aesar) in phosphate-buffered saline (PBS) for 15 min at room temperature or in 100% methanol (A412-4, Fisher Scientific) for 5 min at –20°C. The primary antibodies used for IF are listed in *Source data 4*. All staining condition such as fixation condition and dilution of the antibodies can be found in the source data of each figure. After blocking with 5% normal serum that are matched with the species used to raise secondary antibodies (005-000-121 or 017-000-121, Jackson ImmunoResearch) in IF buffer (3% bovine serum albumin (BP9703100, Fisher Scientific), 0.02% sodium azide (BDH7465-2, VWR International), and 0.1% NP-40 in PBS) for 30 min at room temperature, cells were incubated with primary antibody in IF buffer for at least 3 hr at room temperature, followed by rinsing with IF buffer five times. The samples were then incubated with fluorescent-labeled secondary antibody (listed below) in IF buffer for 1 hr at room temperature, followed by rinsing with IF buffer five times. After nuclear staining with 4',6-diamidino-2-phenylindole (DAPI) (40043, Biotium) in IF buffer at a final concentration of 0.5 µg/ml, coverslips were mounted with Fluoromount-G (0100-01, SouthernBiotech) onto glass slides (3050002, Epredia). Images were acquired on an Everest deconvolution workstation (Intelligent Imaging Innovations) equipped with a Zeiss Axio Imager Z1 microscope and a CoolSnap HQ cooled CCD camera (Roper Scientific). A 40× NA1.3 Plan-Apochromat objective lens (420762-9800, Zeiss) was used for ciliation assays, and a 63× NA1.4 Plan-Apochromat objective lens (420780-9900, Zeiss) was used for other analyses.

For ciliation assays, cells were plated into a 6-well plate at a density of $2 \times 10^5$ cells/well and grown for 66 hr. Cells were serum starved for 24 hr unless otherwise indicated and fixed in 4% PFA. In the experiments presented in *Figure 3A*, the cells were incubated in serum-free media for 12, 24, 48, 72, or 96 hr before fixation. After the blocking step, cells were stained with anti-ARL13B (17711-1-AP, Proteintech), anti-CEP170 (41–3200, Invitrogen), and anti-acetylated tubulin (Ac-Tub) antibodies (T7451, Sigma), washed, then stained with anti-rabbit Alexa Fluor 488 (711-545-152, Jackson ImmunoResearch), goat anti-mouse IgG1-Alexa Fluor 568 (A-21124, Invitrogen), and goat anti-mouse IgG2b Alexa Fluor 647 (A-21242, Invitrogen). All the images were captured by focusing CEP170 without looking at a channel of the ciliary proteins to avoid selecting specific area based on the percentage of ciliated cells. The structures extending from the centrosome and positive for ARL13B with the length

of more than 1 μm was counted as primary cilia. At least six images from different fields per sample were captured for typical analysis. Typically, at least 200 cells were analyzed per experiment. Exact number of cells that we analyzed in each sample can be found in the Source Data of corresponding figures. The percentage of ciliated cells were manually counted using the SlideBook software (Intelligent Imaging Innovations).

For ciliary vesicle recruitment assays, cells were plated into a 6-well plate at a density of 2 × $10^5$ cells/well, grown for 66 hr (without serum starvation), and fixed in 4% PFA. After the blocking step, cells were stained with anti-RAB34 (27435-1-AP, Proteintech), anti-Myosin Va (sc-365986, Santa Cruz), and anti-CEP170 (to mark centriole) antibodies (41–3200, Invitrogen), washed, and then stained with goat anti-mouse IgG2a Alexa Fluor 488 (A-21131, Proteintech), goat anti-rabbit Alexa Fluor 568 (A10042, Invitrogen), and goat anti-mouse IgG1 Alexa Fluor 647 (A-21240, Invitrogen). All the images were captured by focusing CEP170 without looking at a channel of the vesicle markers to avoid selecting specific area based on the percentage of the vesicle positive centrioles. At least eight images from different fields per sample were captured for typical analysis. Typically, at least 50 cells were analyzed per experiment. Exact number of cells that we analyzed in each sample can be found in the Source Data of corresponding figures.

For CP110 removal assays, cells were plated into a 6-well plate at a density of 2 × $10^5$ cells/well and grown for 66 hr. Cells were serum starved for 24 hr in 100% methanol. After the blocking step, cells were stained with anti-CP110 (12780-1-AP, Proteintech), anti-FOP (H00011116-M01, Abnova) (to mark both mother and daughter centrioles), and anti-CEP164 (sc-515403, Santa Cruz) (to mark the mother centriole) antibodies, washed, then stained with anti-rabbit Alexa Fluor 488 (711-545-152, Jackson ImmunoResearch), goat anti-mouse IgG2a-Alexa Fluor 568 (A-21134, Invitrogen), and goat anti-mouse IgG2b Alexa Fluor 647 (A-21242, Invitrogen). All the images were captured by focusing FOP without looking at a channel of the other centriolar proteins to avoid selecting specific area based on the percentage of CP110-positive centrioles. CP110 localizing to both mother and daughter centrioles (as judged by colocalization with FOP) were counted as two dots, and CP110 localizing only to daughter centriole (as judged by no colocalization with CEP164) was counted as a one dot. Exact number of cells that we analyzed in each sample can be found in the Source Data of corresponding figures.

For structured illumination microscopy, cells were grown on 18 mm square coverslips with the thickness of 0.17 mm (474030-9000-000, Zeiss), fixed, and stained as described above. DAPI staining was not included for the structured illumination samples. Coverslips were mounted with SlowFade Gold Antifade Reagent (S36936, Life Technologies). Images were acquired on a DeltaVision OMX V4 system equipped with a 100×/1.40 NA UPLANSAPO100XO objective lens (Olympus), and 488 nm (100 mW), 561 nm (100 mW), and 642 nm (300 mW) Coherent Sapphire solid state lasers and Evolve 512 EMCCD cameras (Photometrics). Image stacks of 2 μm z-steps were taken in 0.125 μm increments to ensure Nyquist sampling. Images were then computationally reconstructed and subjected to image registration by using SoftWoRx 6.5.1 software.

Secondary antibodies used for IF were donkey anti-rabbit Alexa Fluor 488 (711-545-152, Jackson ImmunoResearch), donkey anti-Chicken IgY Alexa Fluor 488 (703-545-155, Jackson ImmunoResearch), donkey anti-mouse IgG DyLight 488 (715-485-150, Jackson ImmunoResearch), goat anti-mouse IgG2a Alexa Fluor 488 (A-21131, Thermo Fisher Scientific), goat anti-mouse IgG$_1$ Alexa Fluor 488 (A-21121, Thermo Fisher Scientific), donkey anti-rabbit IgG Alexa Fluor 568 (A10042, Thermo Fisher Scientific), goat anti-mouse IgG2a-Alexa Fluor 568 (A-21134, Thermo Fisher Scientific), goat anti-mouse IgG1-Alexa568 (A-21124, Thermo Fisher Scientific), goat anti-mouse IgG2b Alexa Fluor 647 (A-21242, Thermo Fisher Scientific), goat anti-mouse IgG1 Alexa Fluor 647 (A-21240, Thermo Fisher Scientific), and donkey anti-rabbit IgG Alexa Fluor 647 (711-605-152, Jackson ImmunoResearch).

## Mice

*Ncs1*$^{-/-}$ mice in a C57BL/6J background were originally generated by the lab of Olaf Pongs (*Hermainski, 2012*) and the strategy for the gene targeting was previously described (*Ng et al., 2016*). Briefly, the 129 strain-derived R1 embryonic stem cells carrying the targeting cassette was injected into C57BL/6J blastocysts. The resulting *Ncs1*$^{-/-}$ mice, which lack exons 4–7 of *Ncs1*, were backcrossed to C57BL/6J over 10 generations. The backcrossed mice were re-derived and maintained at the Toronto Centre for Phenogenomics until they were transferred to Stanford University.

All mice were maintained under specific pathogen-free conditions at the Stanford animal care facility. All experiments were approved by Administrative Panel on Laboratory Animal Care at Stanford University (Institutional Animal Care and Use Committee protocol number: 28556).

The primers used for genotyping PCR are *Ncs1*_genotyping-F: 5′-GTCCACCCATACCAATCACT -3′, *Ncs1*_genotyping_WT-R: 5′-ACAGAGAATCCAAAGCCAGC-3′, *Ncs1*_genotyping_KO-R: 5′- TTGT GCTGGAGAAGGGAGAG-3′. The bands observed by PCR amplifications are 398 and 514 bp for wild-type and knockout mice, respectively.

The term 'littermate controls' used in this paper means that the mice were born from the same mother on the same day and were housed in the same cage as the test animals throughout the life.

## Assessment of viability of $Ncs1^{-/-}$ mice

To test viability of $Ncs1^{-/-}$ mice, $Ncs1^{+/-}$ female and male mice were mated, and genotype of the offspring was examined by genomic PCR using the genotyping PCR primers described above at P21.

## Body weight measurement

The body weight of male or female $Ncs1^{-/-}$ mice and their littermate controls were measured weekly between 9 am and 12 pm. The statistics was obtained through two-way ANOVA with Tukey's multiple comparisons test. All the raw data can be available in *Figure 8—source data 1*.

## Measurement of fat weight

Twenty-week-old $Ncs1^{-/-}$ and their litter mate $Ncs1^{+/-}$ mice were anesthetized with isoflurane and euthanized by cervical dissociation. Inguinal or epididymal fat was then dissected out from the mice and were measured on a scale.

## Preparation, staining, and imaging of the tissue samples

Six- to eight-week-old $Ncs1^{-/-}$ or their litter mate control animals were first anesthetized with 3% isoflurane (Fluriso, Bet-one) at a delivery rate of 1 l/min. Complete anesthesia was confirmed by checking toe pinch reflex, and the animal was kept anesthetized throughout the procedure using a face mask that is connected to the anesthesia machine (VetEquip). Following exposure of the heart, an incision was made in the right atrium. Next, 27G½ gage needle (305109, BD) connected to a 20-ml syringe (302830, BD) was inserted into the left ventricle to transcardially perfuse the animal with 20 ml of PBS followed by 1.5 ml/g (~35 ml) of 4% (vol/vol) PFA (15710, Electron Microscopy Sciences). Note that the transcardiac perfusion of 4% PFA is critical to preserve the sample to visualize primary cilia in tissues. The fixed tissues were dissected out and post-fixed in 20 ml of 100% methanol at −20°C for 20 hr. We found that the post-fixation in methanol is critical for Ncs1 visualization in tissues likely through washing out the PFA from the tissue, since over-fixation of the samples in PFA greatly diminished the centrosomal signal of Ncs1 in monolayer cultured cells (data not shown). The post-fixed tissues were then submerged in graded concentration (10–20–30% (wt/vol)) of sucrose (S9378, Sigma-Aldrich) in PBS at 4°C until the tissue sunk in each solution to cryoprotect the samples. The tissues were then embedded into OCT compound (4583, Tissue-Tek). Cryosections (typically 7–10 µm thickness) were created on a Cryostat (3050S, Leica) and the sliced tissues were collected on adhesive microscope slides (16005-110, VWR). Samples were immunostained using the same procedure as the one used for wide-field microscopy experiments. The stained samples were imaged on the Marianas SDC spinning disk microscope (Intelligent Imaging Innovations) equipped with Cascade 1K camera (photometrics) and CSU22 confocal scanner unit (Yokogawa). A 63× NA1.4 Plan-Apochromat objective lens (420781-9910-000, Zeiss) was used to acquire images. Typically, image stacks of 10–20 µm z-steps were taken in 0.5 µm increments.

## HE stains

20-, 30-, or 50-week-old $Ncs1^{-/-}$ mice and their littermate controls were first fixed by transcardiac perfusion of 4% PFA as described above and post-fixed in 4% PFA at 4°C for 72 hr. Tissues were then processed, embedded in paraffin blocks, sectioned on a microtome, and stained with hematoxylin and eosin by standard techniques. Optimal number of tile pictures was obtained and stitched together via Keyence BZ-X710 fluorescent microscope.

## Isolation of hippocampal neurons

Hippocampus was dissected out from E18.5 mice, which were developed from $Ncs1^{+/-}$ female mouse crossed with $Ncs1^{+/-}$ male mice. The dissected hippocampus was dissociated by incubating the tissue in calcium magnesium-free (CMF)-HBSS media (14175095, Gibco) supplemented with 10 mM HEPES (15630080, Gibco) containing 0.05% trypsin (15400-054, Gibco) at 37°C for 20 min. After washing the trypsinized tissue three times with 500 µl of CMF-HBSS containing 10 mM HEPES, the tissue was triturated with a fire polished Pasteur pipette. The dissociated cells were then plated on a 12-mm round coverslip (12-545-81, Fisher Scientific) coated with poly-D-lysine at a density of 60,000 cells per 24-well plate (930186, Thermo Scientific). The cells were grown in 500 µl of the Neurobasal Medium (21103049, Gibco) supplemented with 1× B27 (17504044, Gibco), 1× GlutaMax, 100 U/ml penicillin–streptomycin, and 10% horse serum (16050130, Gibco). Twenty-four hours after plating, the media were replaced with the Neurobasal Medium media supplemented with 1× B27, 1× GlutaMax, and 100 U/ml penicillin–streptomycin. The genotype of the neurons was confirmed by genotyping PCR using the genotyping PCR primers described above.

## Preparation of MEF

MEFs were prepared from E13.5 mice embryos, which were developed from $Ncs1^{+/-}$ female mice crossed with $Ncs1^{+/-}$ male mice. After removing innards from the embryo, the remaining was minced with a razor blade (55411-050, VWR). The minced tissues were dissociated using 2 ml 0.05% trypsin/EDTA (25300-054, Gibco) for 20 min at 37°C, followed by neutralization of trypsin by adding 4 ml of MEF media (DMEM high glucose (11995073, Gibco), 10% FBS (100-106, Gemini), 1× GlutaMax (35050-079, Thermo Fisher Scientific), and 100 U/ml penicillin–streptomycin (15140163, Thermo Fisher Scientific)) containing 100 µg DNase I (LS002006, Worthington). Cells were then pelleted down, re-suspended in 15 ml of MEF media and plated into a T75 flask. The genotype of the MEFs was confirmed by genotyping PCR using the genotyping PCR primers described above. All experiments were performed with the cells that were passaged no more than three times.

## Immunoblotting of the tissue lysate

A 7-week-old $Ncs1^{-/-}$ and a 6-week-old $Ncs1^{+/+}$ mouse (not a littermate control) were anesthetized with isoflurane and euthanized by cervical dissociation. Tissues were quickly dissected out and minced with a razor blade (55411-050, VWR). The minced tissue is lysed in tissue lysis buffer (50 mM Tris–HCl [pH 7.5], 150 mM NaCl, and 1% NP-40 (11332473001, Roche Applied Science)) for 15 min. Following clarification of the lysate at centrifugation at 15,000 rpm (21,000 × $g$) for 15 min at 4°C, the concentration of the supernatant was measured by Bradford assay as previously described (see procedure B step 8 in *Kanie and Jackson, 2018*). The lysate was mixed to prepare a sample containing 4 mg/ml lysate, 1× LDS buffer, and 2.5% 2-mercaptoethanol. 50 µg (for NCS1 blot) or 12 µg (for other proteins) were loaded onto NuPAGE Novex 4–12% Bis-Tris protein gels. Western blot was performed as described above and the fluorescent signal was detected on an Odyssey CLx Imaging System (LI-COR).

## Experimental replicates

The term 'replicates' used in this paper indicate that the same cell lines were plated at different dates for each experiment. In most cases, cell lines were thawed from liquid nitrogen at different dates and immunostaining was performed at different dates among the replicates.

## Quantification of fluorescent intensity and statistical analysis

### Fluorescent intensity measurement

The fluorescent intensity was measured with 16-bit TIFF multi-color stack images acquired at 63× magnification (NA1.4) by using ImageJ software. To measure the fluorescent intensity of centrosomal proteins, channels containing CEP170 and the protein of interest (POI) were individually extracted into separate images. A rolling ball background subtraction with a rolling ball radius of 5 pixels was implemented for both CEP170 and the POI to perform local background subtraction. The mask for both CEP170 and the POI was created by setting the lower threshold to the minimum level that covers only centrosome. Each mask was then combined by converting the two masks to a stack followed by z-projection and then dilating the mask until the two masks are merged. After eroding the dilated masks

several times, the fluorescent intensity of the POI was measured via 'analyze particles' command with optimal size and circularity. The size and circularity are optimized for individual POI to detect most of the centrosome in the image without capturing non-centrosomal structure. Outliers (likely non-centrosomal structure) were then excluded from the data using the ROUT method with a false discovery rate of 1% using GraphPad Prism 9 software. Fluorescent intensity of ciliary proteins was measured similar to centrosomal proteins but with several modifications. Mask was created for ciliary proteins by setting the lower threshold to the minimum level that covers only cilia. The size and circularity are optimized for individual POI to detect only cilia without capturing non-ciliary structure. Macros used for the intensity measurement are available from 'Source Data 2—Macro for measuring fluorescent intensity of centrosomal proteins' and 'Source Data 3—Macro for measuring fluorescent intensity of ciliary proteins' in an accompanying paper (*Kanie et al., 2025*).

To test whether the difference in the signal intensity is statistically different between control and test samples, the intensity measured through the described method was compared between control and test samples using nested one-way ANOVA with Dunnett's multiple comparisons test or nested t-test if there are more than two replicates. In case, there are less than three replicates, the statistical test was not performed in a single experiment, as the signal intensity is affected slightly by staining procedure and statistical significance is affected largely by the number of cells examined. For example, we saw statistical significance in the signal intensity with the same samples that are stained independently if we analyze large number of the cells (more than 100 cells). Instead, we confirmed the same tendency in the change of fluorescent intensity in the test samples across two replicates.

## Statistical analysis for ciliation, preciliary vesicle recruitment, and CP110 removal assay

For ciliation, preciliary vesicle recruitment, and CP110 removal assay, the number of ciliated cells from the indicated number of replicates was compared between control (sgGFP or sgSafe) and the test samples using Welch's *t*-test. The exact number of samples and replicated are indicated in the Source Data of the corresponding figures.

For all the statistics used in this paper, asterisks denote $*0.01 \leq p < 0.05$, $**p < 0.01$, $***p < 0.001$, n.s.: not significant. All the statistical significance was calculated by using GraphPad Prism 9 software.

## Protein structural prediction using AlphaFold

The structural predictions shown in *Figures 2O and 5G* and *Figure 2—figure supplement 3* were calculated using a local installation of AlphaFold multimer v2.1 (*Jumper et al., 2021*; *Evans et al., 2022*). Sequences of *H. sapiens* NCS1, CEP89, CEP15, SCLT1, and KIZ were used as inputs for the structure predictions. In *Figure 5G*, the crystal structure of NCS1 (PDB ID: 6QI4) was super-imposed on the predicted structural model of NCS1-CEP89 to pinpoint the calcium-binding sites. PyMOL v. 2.5 (Schrodinger LLC, https://pymol.org) was used to prepare figures of protein structures.

## Materials availability statement

All the newly created materials used in this paper including plasmids and stable cell lines are readily available from the corresponding authors (Tomoharu-Kanie@ouhsc.edu or pjackson@stanford.edu) upon request.

## Acknowledgements

We thank Drs. Albert Wong and John Georgiou for providing *Ncs1* knockout mice. We thank Dr. Jonathan Mulholland for technical advice on the 3D-SIM experiments. We thank Mr. John Perrino for technical support for sample preparation for the electron microscopy experiments. We thank members of the Jackson lab for helpful discussion and advice. 3D-SIM experiments were performed at the Stanford Cell Sciences Imaging Facility and were supported by Award Number 1S10OD01227601 from the National Center for Research Resources (NCRR). Electron microscopy observation was performed at the Stanford Cell Sciences Imaging Facility and was supported by National Institutes of Health (NIH) S10 Award Number 1S10OD028536-01, titled 'OneView 4kX4k sCMOS camera for transmission electron microscopy applications'. The cell authentication service performed by MTCRO-COBRE Cell line authentication core of the University of Oklahoma Health Science Center was supported

partly P20GM103639 and National Cancer Institute Grant P30CA225520 of NIH. This project was supported by funds from the Baxter Laboratory for Stem Cell Research, the Stanford Department of Research, the Stanford Cancer Center, NIH grants R01GM114276 and R01GM121565 to PKJ, NIH grant P20GM103447 and 1R35GM151013 to TK.

## Additional information

### Funding

| Funder | Grant reference number | Author |
| --- | --- | --- |
| National Institute of General Medical Sciences | P20GM103447 | Tomoharu Kanie |
| National Institute of General Medical Sciences | 1R35GM151013 | Tomoharu Kanie |
| National Institute of General Medical Sciences | R01GM114276 | Peter K Jackson |
| National Institute of General Medical Sciences | R01GM121565 | Peter K Jackson |

The funders had no role in study design, data collection and interpretation, or the decision to submit the work for publication.

### Author contributions

Tomoharu Kanie, Conceptualization, Resources, Data curation, Formal analysis, Supervision, Funding acquisition, Validation, Investigation, Visualization, Methodology, Writing – original draft, Project administration, Writing – review and editing; Roy Ng, Keene L Abbott, Data curation, Investigation, Writing – review and editing; Niaj Mohammad Tanvir, Investigation, Methodology; Esben Lorentzen, Investigation, Methodology, Writing – review and editing; Olaf Pongs, Resources, Writing – review and editing; Peter K Jackson, Conceptualization, Resources, Supervision, Funding acquisition, Investigation, Methodology, Project administration, Writing – review and editing

### Author ORCIDs

Tomoharu Kanie ⓘ https://orcid.org/0000-0002-2084-1451
Keene L Abbott ⓘ https://orcid.org/0000-0002-6166-704X
Esben Lorentzen ⓘ https://orcid.org/0000-0001-6493-7220

### Ethics

All mice were maintained under specific pathogen-free conditions at the Stanford animal care facility. All experiments were approved by Administrative Panel on Laboratory Animal Care at Stanford University (Institutional Animal Care and Use Committee protocol number: 28556) and were performed in strict accordance with their guidelines.

### Decision letter and Author response

Decision letter https://doi.org/10.7554/eLife.85998.sa1
Author response https://doi.org/10.7554/eLife.85998.sa2

## Additional files

### Supplementary files

MDAR checklist

Source data 1. Primers used for genomic PCR and for generating sgRNA vectors.

Source data 2. The list of mouse embryonic fibroblasts used in this paper.

Source data 3. The list of hippocampal neurons used in this paper.

Source data 4. The list of antibodies used in this paper.

Source data 5. The list of cell lines used in this paper.

Source data 6. Uncropped images of the immunoblot with label.

Source data 7. Summary of CRISPR knockout cells.

## Data availability

All data generated or analyzed during this study are included in the manuscript and supporting file. Source Data files have been provided for corresponding figures.

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
