## [Editor Report]

The identification of NCS1 as a distal appendage protein that captures preciliary vesicles has fundamental implications for understanding the early steps of ciliary assembly, furthering also a broader understanding of NCS1. Prior to this work, studies of NCS1 were focused on its roles in neurotransmission, but now must be considered in a larger context. The investigators used a variety of state-of-the-art methodologies to arrive at compelling conclusions. This work will be of relevance to cell biologists, especially those studying ciliary assembly, as well as human geneticists with an interest in cilia-related pathologies and neurobiologists studying NCS1.

---

## [Decision Letter]

**Decision letter after peer review:**

Thank you for submitting your article "Myristoylated Neuronal Calcium Sensor-1 captures the ciliary vesicle at distal appendages" for consideration by *eLife*. Your article has been reviewed by 3 peer reviewers including Gregory J Pazour as the Reviewing Editor and Reviewer #1, and the evaluation has been overseen by Piali Sengupta as the Senior Editor.

The reviewers have discussed their reviews with one another, and the Reviewing Editor has drafted this to help you prepare a revised submission. As you can see from the reviews detailed below, the work was well received. However, all reviewers had suggestions for clarifying and improving the study that you should consider in your resubmission. I think these comments should be able to be addressed by text changes and no additional experiments are required although you may want to consider adding data as suggested.

*Reviewer #1 (Recommendations for the authors):*

The mouse discussion needs to be more nuanced. "A series of previous mouse genetic studies showed that the loss of ciliary function in mice result in partially penetrant pre-weaning lethality, obesity, retinal degeneration, and male infertility (Nishimura et al., 2004) (Ding et al., 2020; Fath et al., 2005; Mykytyn et al., 2004)." The mouse phenotypes resulting from ciliary defects are much broader than this. Furthermore, the phenotype depends greatly on whether cilia are absent or only reduced and what components are defective. I suggest that authors focus their comparisons on mouse lines with defects in distal appendages in order to understand whether NCS1 is unusual. I suspect that it is probably similar to other distal appendage mutants.

Other points

Why don't the numbers add up in Figure 3D? 22/41 and 16/41.

What is the evidence that Rab34 is not the NCS1-redundant factor that is sought in Figure 4?

The tissue staining in Figure 7 is not convincing. Except for the hypothalamus, I don't see a difference between the control and the knockout tissue. Better images should be included, or this data should be removed.

*Reviewer #2 (Recommendations for the authors):*

It would be good to correct the manuscript for typos, singular versus plural and grammatical mistakes throughout.

*Reviewer #3 (Recommendations for the authors):*

– I am confused about the use of ciliary vesicles to describe the membrane docking affected by NCS1/CEP89. The ciliary vesicle or CV stage is classically associated with a larger membrane cap covering the mother centriole distal end, which I believe may be referred to here as the fused vesicle based on TEM studies reported (Figure 3D). Upstream of the CV is thought to be the docking of small vesicles previously referred to as preciliary vesicles or distal appendage vesicles in several other reports. This can be confusing to readers in light of the authors' statement that this work provides the 'first known mechanism for how the distal appendages recruit the ciliary vesicles', which is a strong statement when considering other reports have described distal appendage protein interactions with membrane-associated factors associated with early ciliogenesis processes.

– The mouse knockout studies are more preliminary in nature given additional experimental options that need to be explored to fully conclude the essential nature of NCS1. One suggestion would be to combine Figures 7 and 8 to focus on the key findings from the mouse studies while placing some data in supplemental.

– The requirement for TTBK2 in NCS1 distal appendage localization is interesting. Can the authors rule out that TTBK2 phosphorylation of NCS1 is not important for distal appendage localization?

– Were double knockouts of C3ORF14 and NCS1 considered to see if there is more than an additive effect on ciliogenesis disruption and possible compensatory effects?

– Investigating NCS1 depletion/knockout in cells thought to use the extracellular pathway such as IMCD3 cells would be interesting to determine if this can explain differences in ciliogenesis effects observed in mice/neurons.

– Did the authors check to see if cilia length was affected in NCS1 knockouts? Related in Figure 7—figure supplement 2 B ependymal cilia seem shorter or affected in NCS-/- cells, and were not commented on in the text.

– Is the image of CEP83 and NCS1 in Figure 2G representative, of some obvious overlap of signals and others that do not overlap? A top view image for CEP89 and NCS1 localization in Figure 2G would be helpful to show this colocalization relationship better.

---

## [Author Response]

Essential revisions:Reviewer #1 (Recommendations for the authors):The mouse discussion needs to be more nuanced. "A series of previous mouse genetic studies showed that the loss of ciliary function in mice result in partially penetrant pre-weaning lethality, obesity, retinal degeneration, and male infertility (Nishimura et al., 2004) (Ding et al., 2020; Fath et al., 2005; Mykytyn et al., 2004)." The mouse phenotypes resulting from ciliary defects are much broader than this. Furthermore, the phenotype depends greatly on whether cilia are absent or only reduced and what components are defective. I suggest that authors focus their comparisons on mouse lines with defects in distal appendages in order to understand whether NCS1 is unusual. I suspect that it is probably similar to other distal appendage mutants.

Based on the suggestion, we rewrote the text to better convey the complexity of the phenotypes of ciliopathy mice. We wrote “A series of previous genetic studies in mice showed that the loss of ciliary function results in a variety of disorders ranging from developmental defects, including neural tube defect, skeletal anomalies as well as left-right patterning defects, to obesity, retinal degeneration, cystic kidney diseases, liver fibrosis, and male infertility {Norris, 2012 #166}. The phenotypes found in ciliopathy mouse models greatly vary depending on the timing of gene deletion and which ciliopathy gene is mutated in the model, likely reflecting the differences in the severity of the defects in cilium formation and function, as well as the cell types that the gene mutations affect.”

We also compared the phenotypes of *Ncs1*^-/-^ mice with those of the mice deficient in distal appendage proteins and discussed this in the “Discussion”. We wrote “Mice lacking the distal appendage protein, FBF1 (Zhang et al., 2021) or ANKRD26 (Acs et al., 2015) (Bera et al., 2008), or the distal appendage associated protein CEP19 (Shalata et al., 2013), display morbid obesity with few other ciliopathy-related phenotypes (e.g., preweaning lethality and hydrocephalus in Fbf1^-/-^ and male infertility in Cep19^-/-^ mice). Interestingly, our data reveal that knockouts of each of these genes in RPE1-hTERT cells show a kinetic defect in ciliation, but that the cells eventually catch up to complete cilium formation (Figure 5A and B in (Tomoharu Kanie et al., 2023) for ANKRD26 and FBF1) (Figure 3C of (T. Kanie et al., 2017) for CEP19). This phenotype is almost identical to that observed in CEP89 or NCS1 knockout cells (Figure 3A). This suggests that quantitative defects in cilium formation defect may drive obesity with few other ciliopathy-related defects.”

Other pointsWhy don't the numbers add up in Figure 3D? 22/41 and 16/41

41 cells analyzed for the cells expressing sgGFP include three ciliated cells. 22 of the centrioles did not have vesicle, 16 for the centrioles had vesicles, and 3 of the centrioles had primary cilia as shown in Figure 3E.

What is the evidence that Rab34 is not the NCS1-redundant factor that is sought in Figure 4?

positive ciliary vesicle to the mother centrioles. Because RAB34 knockout cells show severe ciliation defects, stronger than NCS1 knockouts, we assumed additional factors cooperate with NCS1 for RAB34 recruitment. We cannot exclude the possibility that RAB34 may have multiple functions including a NCS1-redundant function. But we are fairly sure that additional distal appendage proteins remain to be found for this function.

The tissue staining in Figure 7 is not convincing. Except for the hypothalamus, I don't see a difference between the control and the knockout tissue. Better images should be included, or this data should be removed.

We appreciate the reviewer for pointing this out. When we submitted this manuscript, we needed to reduce the file size. During the file size reduction, the images in Figure 7 were converted to low quality images. In the revised manuscript, we made sure that the quality of the images is preserved.

Reviewer #2 (Recommendations for the authors):It would be good to correct the manuscript for typos, singular versus plural and grammatical mistakes throughout.

We have gone through the manuscript, and corrected typos and grammatical errors.

Reviewer #3 (Recommendations for the authors):– I am confused about the use of ciliary vesicles to describe the membrane docking affected by NCS1/CEP89. The ciliary vesicle or CV stage is classically associated with a larger membrane cap covering the mother centriole distal end, which I believe may be referred to here as the fused vesicle based on TEM studies reported (Figure 3D). Upstream of the CV is thought to be the docking of small vesicles previously referred to as preciliary vesicles or distal appendage vesicles in several other reports. This can be confusing to readers in light of the authors' statement that this work provides the 'first known mechanism for how the distal appendages recruit the ciliary vesicles', which is a strong statement when considering other reports have described distal appendage protein interactions with membrane-associated factors associated with early ciliogenesis processes.

Thank you very much for this valuable comment. We agree that the term ‘ciliary vesicle’ that we used is confusing, and changed the word to preciliary vesicle, distal appendage vesicle, and centriole-associated vesicle in the text. We also clarified our definition in the discussion.

In terms of our statement “'first known mechanism for how the distal appendages recruit the ciliary vesicles”, some of the membrane-associated proteins that were shown to localize to ciliary vesicle/preciliary vesicle/distal appendage vesicle (e.g., RAB8) may be functionally linked to distal appendage proteins [PMID: 29244804][PMID: 23253480], but none of them were shown to localize to the distal appendages. So, we believe it would not be an overstatement to say that NCS1 is the first distal appendage protein that directly associates with the ciliary vesicle/distal appendage vesicle/preciliary vesicle.

– The mouse knockout studies are more preliminary in nature given additional experimental options that need to be explored to fully conclude the essential nature of NCS1. One suggestion would be to combine Figures 7 and 8 to focus on the key findings from the mouse studies while placing some data in supplemental.

We strongly agree that significant amount of work is needed to understand physiological importance of NCS1 in ciliary function in vivo. However, we think it would be valuable for cilia/NCS1 community to show that Ncs1 localizes to the ciliary base in various tissues (Figure 7A-D), may affect ciliary formation and function (Figure 7E-H), and Ncs1 knockout mice may exhibit ciliopathy related phenotypes (Figure 8), especially given that Ncs1 has been characterized mainly in neurotransmission, neurite growth, and regulation of membrane trafficking in neurons. Future studies will determine whether the neurological phenotypes in the absence of Ncs1 is attributable to ciliary dysfunction.

– The requirement for TTBK2 in NCS1 distal appendage localization is interesting. Can the authors rule out that TTBK2 phosphorylation of NCS1 is not important for distal appendage localization?

It is certainly interesting that TTBK2 affects localization of NCS1 without affecting CEP89 localization. We currently do not know whether NCS1 can be phosphorylated by TTBK2. TTBK2 may affect localization of NCS1 through phosphorylation of other distal appendage proteins, such as CEP89 and CEP83, both of which were shown to be phosphorylated by TTBK2. This warrants future studies.

– Were double knockouts of C3ORF14 and NCS1 considered to see if there is more than an additive effect on ciliogenesis disruption and possible compensatory effects?

We did not test if C3ORF14 compensates the lack of NCS1, because CEP89 knockout cells showed very similar cilium formation defect to NCS1 knockout cells, while the centriolar localization of both NCS1 and C3ORF14 was almost completely lost in CEP89 knockout cells.

– Investigating NCS1 depletion/knockout in cells thought to use the extracellular pathway such as IMCD3 cells would be interesting to determine if this can explain differences in ciliogenesis effects observed in mice/neurons.

We strongly agree that it is important to test the role of NCS1 in cilium formation in mIMCD3 cells, which was shown to use extracellular pathway for their cilia formation. This warrants future studies.

– Did the authors check to see if cilia length was affected in NCS1 knockouts? Related in Figure 7—figure supplement 2 B ependymal cilia seem shorter or affected in NCS-/- cells, and were not commented on in the text.

According to the reviewer’s advice we checked cilia length in CEP89, NCS1, and C3ORF14 knockout cells, and found no significant difference between control and these knockout cells (see new Figure 3—figure supplement 1A).

As the reviewer pointed out, the cilia length in ependymal cells shown in Figure 7—figure supplement 2B looked different between *Ncs1*^+/+^ and *Ncs1*^-/-^ mice. We believe this difference likely comes from the difference in orientation of the tissue sections. In the Figure 7—figure supplement 2B, the brain slice of the *Ncs1*^+/+^ mice was perpendicular to the ventricular surface, whereas the ependymal cells in Ncs1-/- mice were cut diagonally. The purpose of the figure was to show the ciliary base localization of Ncs1, and we did not pay much attention to the orientation of the ependymal cilia. We agree that this figure is confusing, and decided to replace the images (new Figure 7—figure supplement 2B).

To understand whether Ncs1 affects the cilia length in ependymal cells, substantial number of images are required to accurately measure cilia length in tissue sections, as the measurement can be affected by the orientation of the sections. We currently do not have enough images to perform this analysis.

– Is the image of CEP83 and NCS1 in Figure 2G representative, of some obvious overlap of signals and others that do not overlap? A top view image for CEP89 and NCS1 localization in Figure 2G would be helpful to show this colocalization relationship better.

The position of each signal is affected by several factors, including the orientation of the centriole and how the primary/secondary antibodies attach to the target. NCS1 signal seems to be partially overlapped at the right side of the centriole in Figure 2G, but the peak signal of NCS1 was 50 nm away from that of CEP89 when we analyzed the picture using “profile plot” in the Image J. The difference in the distance of the peak signal between CEP89 and NCS1 was 90 nm at the left side of the centriole, confirming that NCS1 is located slightly above CEP89 at both side of the centriole. The same issue can be observed in Figure 2H, where RAB34 and NCS1 signal partially overlaps at the left side of the centriole with minimal signal overlap at the right side of the centriole. This level of asymmetry is difficult to avoid since the perfectly oriented centrioles are rarely found in the microscope slides. Therefore, we analyzed a lot of centrioles both from top and side view and compared with different markers to conclude the position shown in the cartoons of Figure 2D-I.